



# Resolving effects of leaf pigmentation changes and plant residue on the energy balance of winter wheat cultivation in the ORCHIDEE-CROP model

Ke Yu[1], Yang Su[1,2,3], Ronny Lauerwald[2,4], Philippe Ciais[1], Yi Xi[1], Haoran Xu[1], Xianglin Zhang[2], Nicolas Viovy[1], Amie Pickering[5], Marie Collard[6], Daniel S. Goll[1]

[1]Laboratoire des Sciences du Climat et de l'Environnement, LSCE/IPSL, CEA-CNRS-UVSQ, Université Paris-Saclay, Gif-sur-Yvette, 91191, France
[2]UMR ECOSYS, INRAE AgroParisTech, Université Paris-Saclay, Palaiseau, 91120, France
[3]Département d'Informatique, École Normale Supérieure-PSL, Paris, 75005, France
[4]Department Geoscience, Environment & Society-BGEOSYS, Université Libre de Bruxelles, Bruxelles, Belgium
[5]Game & Wildlife Conservation Trust (GWCT), The Allerton Project, Loddington, Leicestershire, United Kingdom
[6]Département durabilité, systèmes et prospectives, Centre wallon de Recherches agronomiques (CRA-W), 6800 Libramont, Belgique

*Correspondence to*: Ke Yu (ke.yu@lsce.ipsl.fr)

Ke Yu (ke.yu@lsce.ipsl.fr), Yang Su (yang.su@ens.fr), Ronny Lauerwald (ronny.lauerwald@inrae.fr), Philippe Ciais (philippe.ciais@lsce.ipsl.fr), Yi Xi (yi.xi@lsce.ipsl.fr), Haoran Xu (haoran.xu@lsce.ipsl.fr), Xianglin Zhang (xianglin.zhang@inrae.fr), Nicolas Viovy (nicolas.viovy@lsce.ipsl.fr), Amie Pickering (apickering@gwct.org.uk), Marie Collard (ma.collard@cra.wallonie.be), Daniel S. Goll (daniel.goll@lsce.ipsl.fr)

**Abstract.** Crop management impacts climate not only through changes in carbon stocks and greenhouse gas budgets, but also through changes in the heat budget. However, the latter aspect is not yet covered by existing cropping system models. The coupling of dedicated crop models with land surface models may be an attempt to quantify those effects, but is hampered by the simplistic representation of surface albedo as a mix of soil albedo and a static vegetation albedo controlled by vegetation cover. Here, we developed ORCHIDEE-CROP, a land surface model integrating the cropping system model STICS, by incorporating time-varying albedo from crop pigmentation during foliar yellowing and post-harvest crop residue soil cover. We further parameterized the effect of crop residues on surface roughness and soil evaporation affecting the heat budget and partitioning between latent and sensible heat fluxes. Using 10 site simulations, we quantified the impacts of these processes on soil temperature, soil moisture, water and heat fluxes of winter wheat crops. Incorporating foliar yellowing and post-harvest residue cover increased surface albedo by an average of 0.07±0.03 during the foliar yellowing period and 0.02±0.02 during the residue cover period, accordingly inducing surface cooling by −0.42±0.65 °C and -1.39±1.07 °C. During each period, sensible heat flux changed by -0.03±2.34 W m⁻² and 1.30±11.52 W m⁻², while latent heat flux decreased by -1.23±1.78 W m⁻² and -3.59±3.90 W m⁻². Spatiotemporal variability in these effects was driven by site-specific



meteorology and soil properties. Simulations of drying climate scenarios reveal that crop residues left on the field can

progressively increase plant available water over multiple years under dry conditions. This study underscores that crop pigmentation and residues significantly modulate surface energy partitioning, and demonstrates the potential of its management for climate mitigation. The refined modelling framework enables simultaneous assessment of the biogeochemical and biophysical impacts of field operations on the Earth System.

## 1 Introduction

Agriculture and climate are interconnected by greenhouse gas, energy and water budgets (IPCC, 2014; Wu et al., 2016; Chen et al., 2024). These interactions create a dual challenge that agricultural systems are both vulnerable to climate change impacts (e.g., droughts, extreme weather) and significant contributors to global emissions. Adaptive management practices are therefore critical for enhancing agricultural resilience and sustainability while simultaneously reducing the anthropogenic disturbances of the environment and climate. To quantify complex spatiotemporal climate impacts of management practices,

coupling land surface models with dedicated crop modules represents a promising strategy. These models are effective tools for simulating biophysical and biogeochemical processes at the agriculture-climate interface, such as crop physiological growth, yield dynamics, and the exchange of carbon, water, and energy between agroecosystems and the atmosphere (Brisson et al., 2003; Fisher et al., 2014; McDermid et al., 2017). By capturing these processes, land surface models provide essential insights for projecting future climate-agriculture feedbacks under varying management scenarios (Arora et al.,

2023; Timlin et al., 2024). However, a key limitation arises from the inconsistency between the diversity and complexity of real-world agricultural management practices and their overly simplistic representations in current land surface models. This gap significantly hinders the ability of land surface models to reliably predict climate impacts, particularly at regional or farm-specific scales where management decisions are highly required (Fisher et al., 2020).

A key example of the limited ability of land surface models in representing biophysical processes of agriculture-climate

interactions is their treatment of crop albedo dynamics.

Surface albedo, as the fraction of incoming solar radiation that is reflected by the soil and crops, plays a critical role in both radiative and non-radiative climate processes (Seneviratne et al., 2018). Crops and their residues exhibit distinct albedo due to specific traits (e.g., leaf pigmentation, canopy structure) and management practices (e.g., tillage, residue retention, irrigation). For example, existing research shows an averaged albedo of 0.16-0.19 for winter wheat (Şerban et al., 2011;

Sieber et al., 2022; Lei et al., 2024), of 0.18-0.20 for soybean (Costa et al., 2007; Lei et al., 2024), and of 0.15-0.18 for corn (Jacobs et al., 1990, Lei et al., 2024). These albedo differences directly affect global climate by changing the radiative forcing at the top of the atmosphere (Stephens et al., 2015). In addition, changes in surface albedo modify the surface energy balance, potentially redistributing energy into latent and sensible heat fluxes as a non-radiative process. However, land surface models generally use simplistic parameterization of static soil and crop albedo. When temporal dynamics are

included, they are typically limited to changes in the fractional coverage of soil and vegetation. (Dalgliesh et al., 2016; Wu



et al., 2016; Hoogenboom et al., 2024). Lacking the mechanisms to represent changes in crop pigmentation, they are unsuited to quantify climate effects of managements or crop breeding which affect crop residue or crop pigmentation (e.g., no tillage, pale crops) (Drewry et al., 2014).

Winter wheat, as a major cereal crop, undergoes distinct phenological stages with changeable crop pigmentation to modulate surface reflectivity (Zhang et al., 2022). During its growth cycle, winter wheat transitions from chlorophyll-dominated green leaves to senescent yellow leaves post-maturity, where reduced chlorophyll levels expose light-reflecting carotenoids (Féret et al., 2017). Additionally, crop residues left on the field post-harvest introduce additional albedo variability compared with bare soil in conventional tillage systems, with the extent of this fluctuation depending on the baseline bare soil albedo and residue management practices (Simão et al., 2021). Generally higher albedo of vegetation than bare soil in most croplands in Europe possibly induces an increase in surface albedo when covering residues on the soil (Pique et al., 2023). Such physiological development of winter wheat and residue coverage thus the impact the climate by surface albedo, offsetting a portion of warming from greenhouse emissions. However, it is difficult for current land surface models to quantify and evaluate these effects because they are generally ignored or parameterized using simplified assumptions (Davin et al., 2014; Seiber et al., 2022; Yu et al., 2024).

This study simulates the biophysical impacts of winter wheat phenology and residue management on surface energy balance through albedo-mediated processes in the ORCHIDEE-CROP land surface model. To achieve this, we improved the model by refining the representation of foliar pigmentation and crop residue effects on albedo dynamics, as well as their influence on soil evaporation and surface roughness based on observations from 10 European cropland sites. Section 2 details the datasets and methodology used for model parameterization. We briefly review the baseline modeling representation of surface albedo, hydrology, roughness, and energy fluxes before introducing our modifications in ORCHIDEE-CROP. These improvements refine albedo dynamics, soil evaporation, and surface roughness, with two simulations for evaluating the albedo-mediated impact of residue retention conducted under both current and drying climate scenarios. Section 3 presents the results of these refinements, while Section 4 analyses the contribution of albedo changes, driven by foliar pigmentation and crop residues, on modulating surface temperature, latent heat, and sensible heat fluxes. We also discuss the potential of crop residue management for climate adaptation and mitigation. Finally, Section 5 summarizes our findings and conclusions. By incorporating albedo dynamics influenced by crop pigmentation and residue management, this study enhances the realism of coupled cropland surface models and demonstrates its impacts on field-level energy balance under present and future idealised climate.

## 2 Data and methods

The improvement of the ORCHIDEE-CROP model is illustrated in Fig. 1 and structured into 6 subsections: (1) Datasets (section 2.1); (2) Introduction of the ORCHIDEE-CROP model (section 2.2); (3) New parameterization of the ORCHIDEE-



CROP model (section 2.3); (4) Model calibration (section 2.4); (5) Model simulations (section 2.5); (6) Model evaluation (section 2.6).

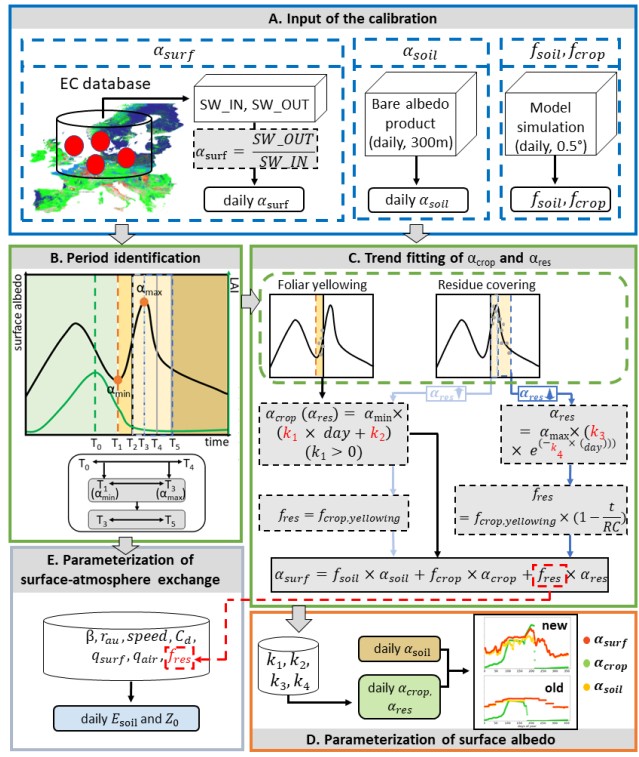

100 **Figure 1** The procedure of parameterization of crop and residue albedo ($\alpha_{surf}$ and $\alpha_{res}$), soil evaporation ($E_{soil}$) and surface roughness ($Z_0$) in the ORCHIDEE-CROP model. $T_{0-5}$ represent the dates of maximum LAI ($T_0$), albedo increase start ($T_1$), harvest ($T_2$) and albedo increase end ($T_3$), 30 days after harvest ($T_4$) and tillage (equals to 90 if no tillage) ($T_5$). $\alpha_{soil}$ and $\alpha_{surf}$ are bare soil albedo ($\alpha_{soil}$) and surface albedo ($\alpha_{surf}$), respectively. $f_{crop}$, $f_{soil}$ and $f_{res}$ are the gridded fractions of crop ($f_{crop}$), soil ($f_{soil}$) and residues ($f_{res}$), respectively. $f_{crop,yellowing}$ is the fraction of crop during the foliar yellowing

105 period. SW_IN and SW_OUT are half-hourly incoming (SW_IN) and outgoing (SW_OUT) solar radiance observed at 7 eddy-covariance towers. $\beta$ is the resistance coefficient; $r_{au}$, speed, $C_d$, $q_{surf}$ and $q_{air}$ are the air density ($r_{au}$), wind speed, drag coefficient ($C_d$), saturated surface air moisture ($q_{surf}$) and specific humidity ($q_{air}$), respectively. $k_1$, $k_2$, $k_3$ and $k_4$ are the fitting parameters.



## 2.1 Datasets

### 2.1.1 Daily surface albedo from site observation

To derive surface albedo ($\alpha_{surf}$, unitless) from site observation, we obtained the half-hourly incoming and outgoing solar radiance (SW_IN and SW_OUT (W m$^{-2}$)) from 2000 to 2020 at 8 wheat cultivated sites (BE-Lon, CH-Oe2, DE-Geb, DE-Kli, DE-RuS, FR-Aur, FR-Gri, FR-Lam) from the Integrated Carbon Observation System (ICOS) Data Portal (Dumont et al., 2023; Schmidt et al., 2023; Buysse et al., 2023; Brut et al., 2023; Brümmer et al., 2023; Tallec et al., 2023; Bernhofer et al., 2023). Data were quality-controlled and gap-filled using uniform methods (Pastorello et al., 2020). We applied a coarse-resolution cloud cover filtering based on the daily moderate-resolution imaging spectroradiometer (MODIS) MO(Y)D04_L2 product (https://ladsweb.modaps.eosdis.nasa.gov/) to exclude days with very low incoming shortwave radiation at the surface which would prevent us from adequately estimating $\alpha_{surf}$. The daily shortwave mid-day $\alpha_{surf}$ was calculated from average values of SW_IN and SW_OUT from 11:00 to 13:00 Central European Summer Time at each site (Lin et al., 2022):

$$\alpha_{surf} = \frac{SW\_OUT}{SW\_IN} \qquad (1)$$

At 6 farms across 2 ClieNFarm sites (UK-Rookery, UK-Winwick, UK-Allerton, BE-AH, BE-BM, BE-LB) where direct radiation measurements were unavailable, we estimated surface albedo at 5-day intervals and 300-m resolution using Sentinel-2 (S2) reflectance data. A 10% cloud cover threshold was applied for quality filtering, and retrievals followed the framework developed by Lin et al. (2023). We only chose years when winter wheat was cultivated for all 10 sites and obtained 33 winter wheat years.

### 2.1.2 Meteorological data from site observation

We utilized the half-hourly meteorological observations from flux towers in the winter wheat years at 8 ICOS sites to force the ORCHIDEE-CROP model. The observations include SW_IN, surface pressure (Pa), near-surface specific humidity (kg kg$^{-1}$), rainfall rate (kg m$^{-2}$ s), snowfall rate (kg m$^{-2}$ s), near-surface air temperature (K), and near-surface eastward and northward wind components (m s$^{-1}$). The FLUXNET_DATA tool was applied (https://forge.ipsl.jussieu.fr/orchidee/wiki/Scripts/FluxnetValidation/DataProcessing) to fill in the flux-towers data gaps using 6-hourly ERA-interim products (Vuichard and Papale, 2015). We then averaged all gap-filled variables to a 6-hour time step to meet the requirement of the model input. At 2 ClieNFarm sites with no flux observations, the meteorological variables from 6-hourly 0.5° CRU-JRA product (v2.4) (Friedlingstein et al, 2022), were utilized as climate forcing data in this model.

In 10-year simulations for the current climate scenario from 2010 to 2020 (section 2.5), the processed climate forcing data from each year obtained from the ICOS Data Portal was used to drive the model. In the drying climate scenario from 2010 to 2020 we utilized ICOS climate forcing data from the year with the lowest annual rainfall at each site to drive the model. The yearly-averaged meteorological variables in the driest year for the drying climate scenario are presented in Table S2.



### 2.1.3 Site management information

Management practices for winter wheat years at all sites (Table S1) were obtained from a crop management database provided by site managers. Key practices, including sowing, harvesting, and tillage dates, were prepared as input for the ORCHIDEE-CROP model.

### 2.1.4 Bare soil albedo from satellite observations

A high-resolution 5-day, 300-m European bare soil albedo ($\alpha_{soil}$, unitless) dataset, derived from 10 m S2 $\alpha_{surf}$ data from 2018 to 2020, was used as forcing for the ORCHIDEE-CROP model (Yu 2025). This dataset demonstrated sufficient skills in resolving daily soil dynamics at the field scale in European fragmented croplands (Yu et al., in review). The new soil dataset was linearly aggregated to 0.5° and a daily time step to align with the spatiotemporal resolution of this model.

### 2.1.5 LAI from satellite observations

Due to the absence or sporadic leaf area index (LAI, $m^2$ $m^{-2}$) measurements found at ICOS sites (Table S3), we used satellite-based LAI datasets to indicate the continuous winter wheat growth at sites. The 4-day, 500-m LAI product from the MODIS (MCD15A3H) was used to extract the LAI of each site (Myneni et al., 2015). Daily LAI dynamics were derived through linear interpolation combined with a Savitzky-Golay filter (window length=15) to smooth the trend and remove the outliers. The LAI comparison between MODIS product and site measurements is shown in Fig. S2. We used MODIS LAI products but not S2 as the latter lacks data before 2018 to match site observations available.

### 2.2 Introduction of the ORCHIDEE-CROP model

The ORCHIDEE-CROP model is a branch version of the process-based land surface model, ORCHIDEE (Krinner et al., 2005), which incorporates a generic crop phenology and harvest module, along with simplified nitrogen fertilization parameterization based on the process-based STICS formalism (Wu et al., 2016). It simulates biophysical and biogeochemical interactions in croplands, as well as plant productivity and harvested yield. A detailed model description is available in Wu et al. (2016). This study employs a version calibrated on LAI, flowing and harvesting dates, and crop yield for winter wheat and maize as the starting version (https://github.com/yangsssu/ORCHIDEE-CROP.git). The derived climate and management information drive the model as forcing data (sections 2.1.2 and 2.1.3). Winter wheat is the only vegetation type considered in this analysis. To assess the impact of physiological development of winter wheat and residue coverage on surface energy balance and water-energy fluxes, we describe below the key processes governing surface energy balance (section 2.2.1), $\alpha_{surf}$ dynamics (section 2.2.2), hydrology (section 2.2.3), the surface roughness (section 2.2.4), as well as crop growth and development (section 2.2.5).





### 2.2.1 Surface energy balance

The ORCHIDEE-CROP model simulates the surface energy budget by:

$$R_n = LE + H + G \tag{2}$$

Where the net radiation ($R_n$, W m$^{-2}$) governs land-atmosphere water and heat exchange, driven by the partitioning of LE, sensible heat fluxes (H, W m$^{-2}$) and soil heat fluxes (G, W m$^{-2}$). $R_n$ is determined by the net short-wave radiation budget ($S_n$, W m$^{-2}$) and the net long-wave radiation budget ($L_n$, W m$^{-2}$) (Ducoudré et al., 1993).

$$R_n = S_n + L_n = (1 - \alpha_{surf}) \times SW\_IN \downarrow + L \downarrow - L \uparrow \tag{3}$$

Where L↓ is the down-welling longwave radiation (a model input variable). L↑ is the upwelling longwave radiation emitted by the land surface, which is estimated by surface temperature ($T_{surf}$, K) (a model input variable) based on Stefan-Boltzmann law.

Since $\alpha_{surf}$ directly influences $R_n$, we next describe its formulation in the ORCHIDEE-CROP model.

### 180 2.2.2 Surface albedo dynamics

The model calculates $\alpha_{surf}$ in step t for both the visible and near-infrared domains by the area-weighted sum of foliar albedo ($\alpha_{crop}$, unitless) and $\alpha_{soil}$:

$$\alpha_{surf}(t) = (1 - f_{crop}(t)) \times \alpha_{soil}(t) + f_{crop}(t) \times \alpha_{crop,constant} \tag{4}$$

Where $f_{crop}$(t) and 1-$f_{crop}$(t) represent the fractions of winter wheat and bare soil covering the surface area of the model

pixel; $\alpha_{crop,constant}$ of winter wheat is set as the default value of 0.1 and 0.3 for visible and near-infrared ranges of $\alpha_{crop}$, respectively. $\alpha_{soil}$(t) is derived from daily $\alpha_{soil}$ dataset (section 2.1.4). Since $\alpha_{surf}$ affects radiation absorption, it influences soil moisture availability and, consequently, soil evaporation (E$_{soil}$).

### 2.2.3 Hydrology

The ORCHIDEE-CROP model computes soil water budget by considering the three reservoirs of canopy interception, snow

packing and 11 soil layers (MacBean et al., 2020). The water fluxes at soil surface include water input to the soil by throughfall, snowmelt and irrigation, and water output from soil by surface runoff, bare soil evaporation and drainage. Here, we mainly described the calculation of E$_{soil}$ and the soil water content (SWC, kg m$^{-2}$) in different soil layers.

The ORCHIDEE-CROP model calculates LE by accounting for snow sublimation, canopy interception and transpiration, E$_{soil}$ and floodplain evaporation. A $\beta$-model is used to compute E$_{soil}$ by inducing a series of resistances.

$$E_{soil}(t) = (1 - \beta_1) \times (1 - \beta_5) \times \beta_4 \times r_{au} \times v \times C_d \times (q_{surf} - q_{air}) \tag{5}$$

Where $\beta_1$, $\beta_4$ and $\beta_5$ represent the surface to atmosphere resistances for snow sublimation ($\beta_1$), soil evaporation ($\beta_4$) and floodplain evaporation ($\beta_5$), respectively; $r_{au}$ (kg m$^{-3}$), $v$ (m s$^{-1}$), $C_d$ (m s$^{-1}$), $q_{surf}$ (kg kg$^{-1}$), and $q_{air}$(kg kg$^{-1}$) are the air density, wind speed, drag coefficient, saturated surface air moisture, and specific humidity, respectively.





Soil moisture dynamics are governed by a diffusive multi-layer soil hydrology scheme with 27 soil layers (up to 2 m), based
on moisture diffusion principles (Richards, 1931; de Rosnay et al., 2002; Wu et al., 2016). It uses the 1D Richards equation
with soil volumetric water content as the state variable in its saturated form, instead of the pressure head (Campoy et al.,
2013).

### 2.2.4 Surface roughness

Surface roughness ($Z_0$, m) plays a critical role in regulating near-surface turbulence and the water-heat exchange process
between the surface and atmosphere (Meier et al., 2022; Chen et al., 2024). $E_{soil}$ depends on $Z_0$, which modulates
atmospheric turbulence and moisture transfer. The ORCHIDEE-CROP model uses the averaged drag coefficients for
momentum and heat for winter wheat to compute the grid-averaged roughness height.

$$Z_0(t) = f_{crop}(t) \times \left(\frac{C_k}{log^{\left(\frac{Z_{tmp}}{max(Height \times Z_{0,h}, Z_{0,bare})}\right)}}\right)^2 \qquad (6)$$

Where $Z_0$ (t) is the daily roughness height over each grid cell. $C_k$ (-) is the von Karman constant, which equals to 0.35. $Z_{tmp}$
(m) represents the maximum height of the first atmospheric layer and is determined by air pressure, air temperature, and air
density. It is the reference height used for flux calculations and is set to a minimum of 10 m above the ground. This ensures
that it remains higher than the canopy height, maintaining the stability of the logarithmic function used in flux calculations.
Height (m) is the winter wheat height which is a time-invariant constant and set to 1.1 m in ORCHIDEE-CROP model. For
comparison, maximum Height of winter wheat during growing seasons measured at 5 sites has an average of 0.97 m (Table
S3). $Z_{0,h}$ equals to 1/16, which is utilized to estimate the roughness height above the canopy. $Z_{0, bare}$ (m) is set as 0.01 m for
roughness length of bare soil. $Z_0$ for the bare soil can be derived by:

$$Z_0(t) = (1 - f_{crop}(t)) \times \left(\frac{C_k}{log^{\left(\frac{Z_{tmp}}{Z_{0,bare}}\right)}}\right)^2 \qquad (7)$$

The ORCHIDEE-CROP model computes a grid effective roughness height ($H_{rough}$, m) for deriving water-heat fluxes. It
combines the zero-plane displacement height ($Z_{dis}$, m) which is an equivalent height for the absorption of momentum, and
the weighted height of vegetation with the maximum $f_{crop}$ ($f_{crop,max}$) for winter wheat in each grid ($H_{ave}$, m):

$$H_{rough} = H_{ave} - Z_{dis} \qquad (8)$$

$$Z_{dis} = H_{ave} \times H_{dis} \qquad (9)$$

$$H_{ave} = H_{ave} + f_{crop,max} \times Height \qquad (10)$$

$H_{dis}$ (m) is used to determine the $Z_{dis}$ based on $H_{ave}$, which equals to 0.75.

### 2.2.5 Crop growth and development

Crop development in the ORCHIDEE-CROP model is based on the crop model STICS (Wu et al., 2016). For winter wheat,
the crop module simulates nine developmental stages of crop growth and grain filling (Fig. 2.1 in Brisson et al., 2008). The





timing and duration of each stage are calculated based on growing degree days, adjusted by limiting functions for photoperiod, vernalization and limited factors (e.g., water and nitrogen). Transitions between stages occur when the
threshold values of growing degree days are reached.

The maturity and harvesting dates were decided by the growing degree days and grain water content in the crop development processes of the ORCHIDEE-CROP model. The threshold of meeting harvesting conditions was calibrated using DWD crop phenology and crop yield datasets from approximately 1000 Germany climate stations (https://opendata.dwd.de/climate_environment/).

**2.3 New parameterization of the ORCHIDEE-CROP model**

Foliar yellowing of winter wheat after maturation and crop residues remaining on the field after harvest affect $\alpha_{surf}$ and turbulent fluxes but are currently not accounted for. As such, the absence of these key processes presents a gap in understanding and modeling the impact of winter wheat on regional energy and water cycles. We enhanced the ORCHIDEE-CROP model by introducing empirical functions describing $\alpha_{surf}$, $\beta_4$ and $Z_0$ dynamics as a function of time, $f_{crop}$ and $f_{res}$
for foliar yellowing and crop residues. In addition, to align the total crop development period (from sowing to harvest) between simulations and observations at each site, the duration between maturity and harvesting in ORCHIDEE-CROP was calibrated using recorded harvesting dates at each site. It means that the durations of foliar yellowing before harvesting vary across all sites in the new model, compared to the default 14 days in the old model.

**2.3.1 Improving albedo dynamics during the foliar yellowing periods**

The foliar yellowing period in this model is defined as starting from the maturity date (nmat) and ending on the harvesting date based on analysis of observed $\alpha_{surf}$ dynamics at 10 sites (section 2.4.1).

We replaced Eq. (4) with an equation which uses time varying $\alpha_{crop}(t)$ instead of a $\alpha_{crop,constant}$:

$$\alpha_{surf}(t) = (1 - f_{crop}(t)) \times \alpha_{soil}(t) + f_{crop}(t) \times \alpha_{crop}(t) \tag{11}$$

To simulate the increase in $\alpha_{crop}$ after maturation and before harvesting we chose the following function:

$$\alpha_{crop}(t) = \alpha_{yellowing,start} \times (k_1 \times t + k_2) \tag{12}$$

Where $\alpha_{yellowing,start}$ is the simulated $\alpha_{crop}$ on the first day during the foliar yellowing period. $k_1$ and $k_2$ represent the fitting parameters that are calibrated at the multi-site scale (see below).

Here, we assumed that $f_{crop}$ remains constant during the foliar yellowing periods ($f_{crop,yellowing}$).

**2.3.2 Improving albedo dynamics during the residue covering periods**

The residue covering period starts from the harvesting date and ends 90 days after harvesting, or earlier if tillage occurs (section 2.4.1). As $\alpha_{surf}$ continuously increases after harvesting due to the light-colored and evenly distributed straws, we used the segmentation function to describe $\alpha_{surf}$ dynamics from increase to decrease during this period.




The $\alpha_{surf}$ at each time step can be represented by the area-weighted sum of $\alpha_{soil}$ and $\alpha_{res}$:

$$\alpha_{surf}(t) = (1 - f_{res}(t)) \times \alpha_{soil}(t) + f_{res}(t) \times \alpha_{res}(t) \tag{13}$$


$$\alpha_{res}(t) = \alpha_{yellowing,start} \times (k_1 \times t + k_2) \quad \text{if } t \in \text{ albedo increase phase;}$$

$$\alpha_{res}(t) = \alpha_{max} \times (k_3 \times e^{-k_4 \times t}) \quad \text{if } t \in \text{ albedo decrease phase; otherwise, } \alpha_{res}(t) = 0 \tag{14}$$

Where $\alpha_{max}$ is the simulated $\alpha_{crop}$ on the last day of the albedo increase phase. $(1 - f_{res}(t))$ represent the daily variation of bare soil fraction.

Here, $f_{res}$ during the albedo increase phase equals to $f_{crop,yellowing}$ and linearly decreases during the albedo decrease stage

from $f_{crop,yellowing}$:

$$f_{res}(t) = f_{crop,yellowing} \text{ if } t \in \text{ albedo increase phase; } f_{res}(t) = f_{crop,yellowing} \times (1 - \frac{t}{RC}) \text{ if } t \in \text{ albedo decrease phase;}$$

$$\text{otherwise, } f_{res}(t) = 0 \tag{15}$$

RC is the maximum residue covering period, either equals to 90 days or the tillage date; $k_3$ and $k_4$ represent the fitting parameters that need to be calibrated at the site scale (see below).

### 2.3.3 Considering the impact of crop residue on soil evaporation during the residue covering periods


Crop residues covering part of soil surface decrease E$_{soil}$ by increasing the soil-to-atmosphere resistance (Horton et al., 1996). The current ORCHIDEE-CROP model does not consider the impact of crop residue on E$_{soil}$. Previous site-scale observations generally show a 20-50 % reduction in E$_{soil}$ due to the presence of wheat residue compared with bare soil (Unger et al., 1986; Lascano, 1996; Ramos et al., 2024). Based on Eq. (5), we assumed a maximum initial soil conductance

($\beta_4$, unitless) reduction of 50 % at the highest $f_{res}$ (Zhao et al., 2020; Raes et al., 2022), with a progressive increase of conductance as daily $f_{res}$ declines during the residue cover period:

$$E_{soil}(t) = (1 - \beta_1(t)) \times (1 - \beta_5(t)) \times (1 - 0.5 \times f_{res}(t)) \times \beta_4(t) \times r_{au}(t) \times v(t) \times C_d(t) \times (q_{surf}(t) - q_{air}(t)) \tag{16}$$

The meanings of all variables are the same as the ones in Eq. (5).

### 2.3.4 Considering the impact of crop residue on surface roughness during the residue covering periods


The initial version of the ORCHIDEE-CROP model assumed that $Z_0$ after harvest equals that of bare soil coverage, omitting that (parts of the) stalks remain. To represent the impact of crop residues on $Z_0$ during the residue covering periods, we assumed that the Height of plant changes from 1.1 m for winter wheat to 0.5 m for crop residues which was found to be optimal for the cutting height for harvest efficiency and soil and water conservation (MacMaster et al., 2000). $\boldsymbol{f_{crop}}$ is

replaced with $\boldsymbol{f_{res}}$. The new $Z_0$ and $\boldsymbol{H_{rough}}$ are therefore computed by:

$$Z_0(t) = f_{res}(t) \times \left(\frac{C_k}{\log^{\left(\frac{Z_{tmp}}{\max(0.5 \times Z_{0,h}, Z_{0,bare})}\right)}}\right)^2 \tag{17}$$



$$H_{rough} = f_{crop,max} \times 0.5 - f_{crop,max} \times 0.5 \times H_{dis} \qquad (18)$$

## 2.4 Model calibration

The parameters of $\alpha_{surf}$ dynamics were calibrated using data from 10 sites with 33 winter wheat site-years. $\alpha_{surf}$ and LAI

observations were applied to identify foliar yellowing and residue covering periods at the available sites (Table S1).

### 2.4.1 Identifying the periods of foliar yellowing and residue covering from site observation

At the site scale, we assumed that foliar yellowing (i.e., $\alpha_{surf}$ increase) begins when maturity is reached due to the absence of observational data supporting a temporal gap between them.

In the model, $\alpha_{surf}$ starts to increase with a delay after maximum LAI is reached and ends with a delay after harvest. Post-

harvest $\alpha_{surf}$ dynamics were divided into two stages: (1) an initial $\alpha_{surf}$ increase driven by light-colored, evenly distributed straw, followed by (2) a $\alpha_{surf}$ decline due to the dominating effects of decomposition of residues.

$\alpha_{surf}$ dynamics were parameterized by identifying critical temporal thresholds. The minimum $\alpha_{surf}$ between the LAI peak and harvest marked the onset of foliar yellowing, which is approximately14.64±10.75 days pre-harvest computed by site observation. This delay is close the assumed delay between maturity date and harvest in ORCHIDEE-CROP of 14 days.  The

maximum $\alpha_{surf}$, constrained to occur in 30 days post-harvest, divided residue-driven $\alpha_{surf}$ into an increase phase (harvest to maximum $\alpha_{surf}$) and a decline phase from post-maximum $\alpha_{surf}$ to tillage. Without tillage, the residue covering period persists for 90 days post-harvest (Kriauciuniene et al., 2012). On average, the maximum $\alpha_{surf}$ occurs 13.36±10.35 days after harvest based on site identification. Due to the lacking information of maximum $\alpha_{surf}$ during residue covering period in the old model, we utilized 15 days after harvesting as the boundary of these two phases in the ORCHIDEE-CROP model at all

sites for simplification.

### 2.4.2 Parameter optimization of albedo dynamics

Daily $\alpha_{surf}$ and $\alpha_{soil}$ observations, as well as and information on area of bare soil exposure were utilized to calibrate the empirical parameters $k_1$, $k_2$, $k_3$ and $k_4$ for $\alpha_{crop}$ and $\alpha_{res}$ (Eqs. 12 and 14). The $\alpha_{yellowing,start}$ (Eq. 12) was replaced by the $\alpha_{surf}$ observation at the beginning of the foliar yellowing period. The simulated $f_{crop}$ from the old model and calculated $f_{res}$

(Eq. 13) were used due to the lack of observation of fraction changes.

Data collected from 10 sites with 28 winter wheat years (N = 485 daily $\alpha_{surf}$ observation) were used to train $k_1$, $k_2$, $k_3$ and $k_4$ in Eqs. 15 and 19. Data from the remaining 5 sites with 5 winter wheat years (N = 179 daily $\alpha_{surf}$ observation) were utilized for testing. Finally, calibrated $k_1$, $k_2$, $k_3$ and $k_4$ were parameterized in the ORCHIDEE-CROP model to simulate the albedo dynamics affected by foliar yellowing and residue covering for all 10 sites.



## 2.5 Model simulations

Simulations were performed for 10 sites (Table S1) with the new ORCHIDEE-CROP model version and with the original version of this model (Table 1) for the years for which observations were available. The experiments aimed to evaluate the impact of the parameterizations of $\alpha_{surf}$, $\beta_4$ and $Z_0$ on water-energy cycles during the foliar yellowing and residue covering periods. In the following, we used ORC-D for the model version before modifications. To test the respective impacts of the 5 model modifications we used 5 different configurations of the improved model. Configuration ORC-AE accounts for changes in $\alpha_{surf}$, $\beta_4$ and $Z_0$ during foliar yellowing and residue covering periods. Configuration ORC-A includes only the modified $\alpha_{surf}$, while ORC-E incorporates adjustments to both $\beta_4$ and $Z_0$. To further disentangle the individual impacts of $\beta_4$ and $Z_0$, ORC-$Z_0$ modifies only the $Z_0$, and ORC- refers to only the modification of the $\beta_4$. Prepared climate forcing data (section 2.1.2), soil texture (sand, silt and clay contents) and crop management information (i.e., sowing date, tillage date, residue covering periods) (section 2.1.3) at each available site were applied to drive the model. Note that all other management practices prescribed in this model (e.g., nitrogen fertilization) were set as default due to the lack of historical implementation records. There is no irrigation consideration in this analysis.

**Table 1** Simulation experiments

| Names of experiments | Surface albedo | Soil conductance | Surface roughness | crop&residue fraction | Test purpose |
|---|---|---|---|---|---|
| ORC-D | no change | no change | no change | no change | - |
| ORC-AE | 1.foliar albedo increases with time during foliar yellowing; 2. residue albedo decreases with time during residue covering | 1.change with new crop fraction during foliar yellowing; 2. change with residue fraction during residue covering; | 1.change with new crop fraction during foliar yellowing; 2. change with residue fraction during residue covering; | 1.crop fraction keeps constant during foliar yellowing; 2. residue fraction decreases with times during residue covering; | combined impacts of surface albedo, soil resistance and surface roughness during two periods |
| ORC-A | 1.foliar albedo increases with time during foliar yellowing; 2. residue albedo decreases with time during residue covering | no change | no change | 1.crop fraction keeps constant during foliar yellowing; 2. residue fraction decreases with times during residue covering; | isolated impact of surface albedo during two periods |





| ORC-E | no change | 1.change with new crop fraction during foliar yellowing; 2. change with residue fraction during residue covering; | 1.change with new crop fraction during foliar yellowing; 2. change with residue fraction during residue covering; | 1.crop fraction keeps constant during foliar yellowing; 2. residue fraction decreases with times during residue covering; | combined impact of soil resistance and surface roughness during two periods |
| --- | --- | --- | --- | --- | --- |
| ORC-β | no change | 1.change with new crop fraction during foliar yellowing; 2. change with residue fraction during residue covering; | no change | 1.crop fraction keeps constant during foliar yellowing; 2. residue fraction decreases with times during residue covering; | isolated impact of soil resistance during two periods |
| ORC-$Z_0$ | no change | no change | 1. change with new crop fraction during foliar yellowing; 2. change with residue fraction during residue covering; | 1. crop fraction keeps constant during foliar yellowing; 2. residue fraction decreases with times during residue covering; | isolated impact of surface roughness during two periods |

*"no change" means variables are the ones from the initial version of the ORCHIDEE-CROP model;

"During both periods" means either during the foliar yellowing or the residue covering period.

In addition, we conducted a 10-year simulation (2010-2020) under current and idealised drying climate scenarios to quantify the cumulative effects of residue soil cover on water and heat fluxes for the new and old models at 6 sites if 10-year climate forcing data is available (Table S2). The meteorological forcing data for both scenarios followed the methodology described in Section 2.1.2, where the current climate scenario used observed annual climate data from ICOS, and the drying climate

scenario was driven by climate forcing from the driest year (the lowest annual rainfall). At each site, the sowing and harvesting dates were the averages of the years where observations were available. The residue covering periods were assumed to persist for 60 days following winter wheat harvest in each year.





### 2.6 Model evaluation

We evaluated the performance of ORCHIDEE-CROP with and without modification against observations of $\alpha_{surf}$ at site scale. The sites used for evaluation were independent from the ones used for calibration of $\alpha_{surf}$.

We computed the coefficient of determination ($R^2$) and root mean square error (RMSE) as quality indicators for simulation-observation comparison of $\alpha_{surf}$ during the site calibration and model simulation processes (Wright 1921, Carbone and Armstrong 1982):


$$R^2 = 1 - \frac{\sum_i^n (y_i - f_i)^2}{\sum_i^n (y_i - \bar{y})^2} \quad\quad (19)$$

$$\text{RMSE} = \sqrt{\sum_i^n \frac{(f_i - y_i)^2}{n}} \qu\quad (20)$$

Where n and i are the number of data and the data i in n in dataset; $y_i$ and $f_i$ are the value of observed data i and the value of fitted data i; y is the mean of the observed data.

With the calibration of the simulation of harvesting date for all sites, we further compared the surface temperature ($T_{surf}$),

LE, H from simulations of the old and new models and observation during the foliar yellowing and residue covering periods.

### 3 Results

### 3.1 Evaluation of surface albedo dynamics in the refined ORCHIDEE-CROP model

The comparison between the predicted (ORC-AE configuration) and observed surface albedo ($\alpha_{surf}$) shows a good performance with an $R^2$ of 0.54 during the foliar yellowing period and of 0.62 during the residue covering period for 5 sites.

The corresponding RMSE has values of 0.04 and 0.03, respectively (Fig. 2). The model underestimates high values, and overestimates low values, which is a typical problem of random forest regression (Belitz et al., 2021).

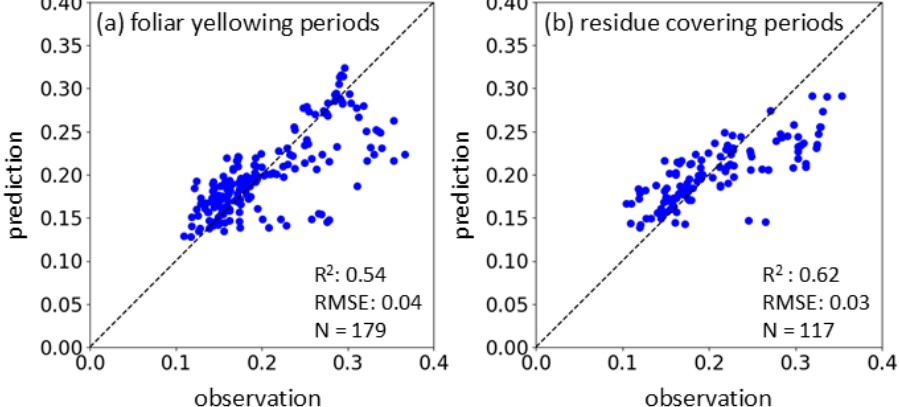

**Figure 2** The comparison of daily surface albedo ($\alpha_{surf}$) predictions with independent observation from 5 sites in Europe during (a) the foliar yellowing and (b) residue covering periods. The dotted black line is the 1:1 line.




ORC-AE captured observed $\alpha_{surf}$ evolution during the yellowing and crop residue periods at all 5 sites. Using FR-Gri as an example (Fig. 3), the initial version of ORCHIDEE-CROP (ORC-D) without considering their effects on $\alpha_{surf}$ dynamics shows substantial biases in $\alpha_{surf}$ during the same period. The results of $\alpha_{surf}$ at all 5 sites can be found in Fig. S4. $\alpha_{surf}$ increases with the development of crops LAI in spring due to the enhanced reflected solar radiation of leaves compared to

darker soil followed by a stable period before crops reach maturity. After the crop has reached maturity, the yellowing of aboveground crop biomass increases $\alpha_{surf}$. This increase is reversed after harvest as dead and dying plant material (i.e., residues) start decomposing. Although the refined model can describe $\alpha_{surf}$ trends in both foliar yellowing and residue covering periods, substantial biases in $\alpha_{surf}$ remain for other periods which were not addressed in this study. This results from inaccuracies of estimating bare soil albedo (Yu et al., in review), from the growth of weeds which are not resolved in

ORCHIDEE-CROP, and from the simplistic representation of vegetation albedo dynamics before maturity is reached.

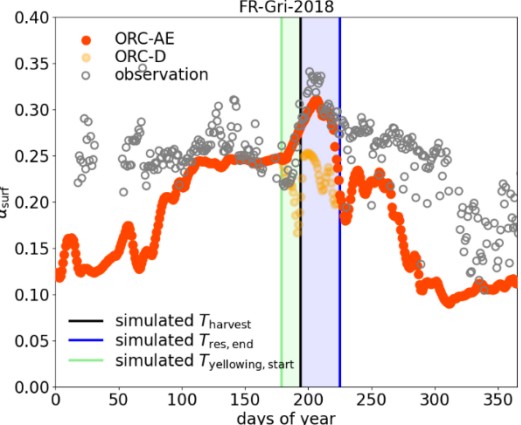

**Figure 3** The comparison of surface albedo ($\alpha_{surf}$) predicted from the old (orange dots) and new (red dots) ORCHIDEE-CROP models and observations at Grignon site in France (gray dots) in 2018. The observed $\alpha_{surf}$ is computed from site eddy covariance measurements through the Integrated Carbon Observation System (ICOS) Data Portal. The green, black and

blue solid lines are the simulated start of the foliar yellowing period (shallow green area), harvesting date and the end of residue covering period (shallow blue area), respectively.

## 3.2 The impact of improved surface albedo, surface roughness and soil conductance on soil evaporation, surface temperature and soil water content

ORC-AE simulates slightly lower soil evaporation ($E_{soil}$) during foliar yellowing and residue covering periods than the standard version of ORCHIDEE-CROP (ORC-D), with averaged decreases of -0.01±0.04 mm d$^{-1}$ and -0.11±0.14 mm d$^{-1}$, respectively (Fig. 4a). The effects of soil conductance ($\beta_4$) and soil roughness($Z_0$) reduced $E_{soil}$ on average by -0.01±0.03




mm/d and -0.09±0.12 mm d$^{-1}$ (ORC-E) during foliar yellowing and residue covering periods, respectively. Impacts on $\alpha_{surf}$ alone cause reductions of respectively -0.01±0.04 mm d$^{-1}$ and -0.06±0.10 mm d$^{-1}$ (ORC-A) for the same periods. The

influence of crop residues on E$_{soil}$ disappears within 40 days after residue coverage (Fig. S5a, Fig. S6a).

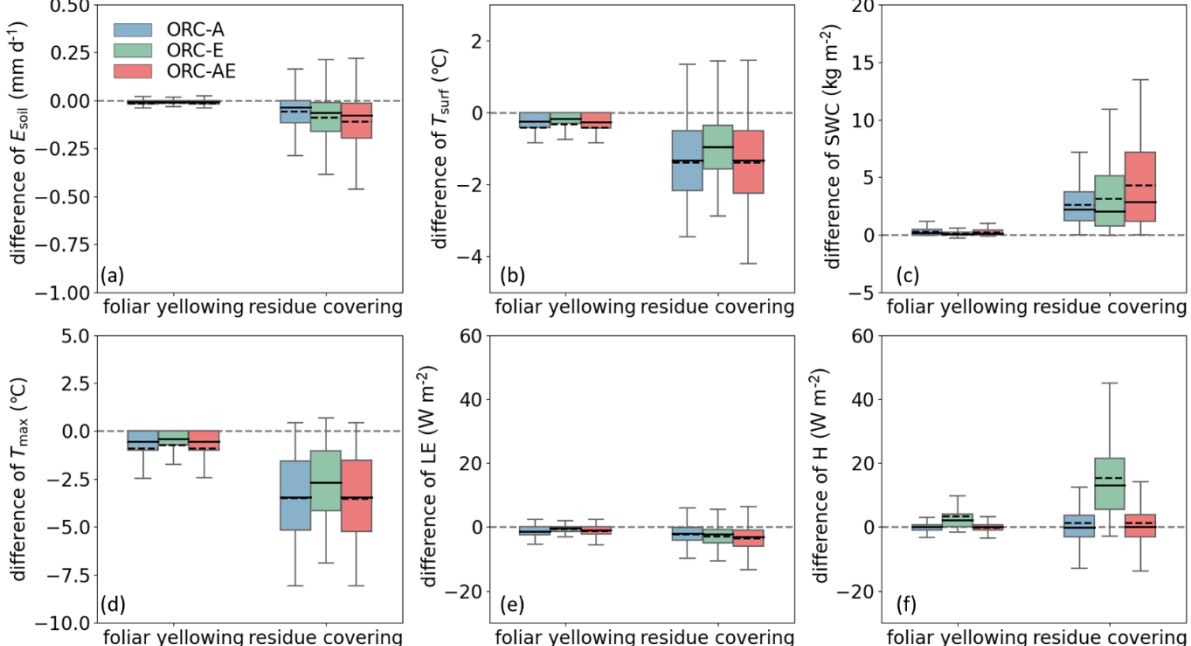

**Figure 4** The average effect of model improvements for yellowing and residues cover on daily soil evaporation ($E_{soil}$), surface temperature ($T_{surf}$), the total soil water content up to 2 m (SWC), the maximum daily surface temperature ($T_{max}$), the latent heat flux (LE) and the sensible heat fluxes (H) across 12 sites. Shown are the simulated differences between three

configurations of the refined model and old model: ORC-A represents the effect of the modified surface albedo ($\alpha_{surf}$), ORC-E shows the effect of refined soil conductance ($\beta_4$) and surface roughness (Z$_0$), and ORC-AE suggests the effect of both modifications. The solid and dotted black lines within each box indicate the mean and median daily values across sites and timepoints, respectively. The box spans the interquartile range, covering the 25th percentile (lower bound) to the 75th percentile (upper bound). Black crossbars at the top and bottom represent the dataset's minimum and maximum values.


The reduced E$_{soil}$ in ORC-AE leads to a minor increase in soil water content (SWC) during the residue covering period among all sites, which is 4.31±3.58 kg m$^{-2}$ (0.66±0.53%) over 2 m soil depth and averaged over the period (Fig. 4c). The combined effect is less than the sum of the changes caused by $\alpha_{surf}$ (2.64±1.72 kg m$^{-2}$ (0.41±0.26%), ORC-A) and by $\beta_4$ and Z$_0$ (3.15±2.89 kg m$^{-2}$ (0.49±0.44%), ORC-E). The water conservation effect persists to the end of year, with gradual

infiltration into deeper soil layers (Fig. S7).

Incorporating changes in $\alpha_{surf}$, $\beta_4$ and Z$_0$ in ORC-AE results in a reduction in surface temperature ($T_{surf}$) during foliar yellowing and residue covering periods with averaged cooling of -0.42±0.65 °C and -1.39±1.07 °C over all sites,



respectively, compared to the ORC-D simulation (Fig. 4b). The factorial simulation experiments show that this cooling effect can be attributed to changes in $\alpha_{surf}$ (ORC-A), $E_{soil}$ and $Z_0$ (ORC-E). Increased $\alpha_{surf}$ alone leads to $T_{surf}$ reductions of -0.41±0.65 °C and -1.37±1.04 °C on average during foliar yellowing and residue covering periods, respectively. Changes in $\beta_4$ and $Z_0$ cause a cooling of -0.31±0.52°C and -0.95±0.88°C for the same periods, respectively. Interaction between $\alpha_{surf}$, $\beta_4$ and $Z_0$ (ORC-AE) lead to dampened overall effect compared to individual effects of these factors on $T_{surf}$. The reduction of $T_{surf}$ tends to weaken over the 60 days of decreasing residue coverage (Fig. S5d, Fig. S6d).

**3.3 The impact of improved surface albedo, surface roughness and soil conductance on latent and sensible heat flux**

Compared with ORC-D simulation, the latent heat flux (LE) in ORC-AE is -1.23±1.78 W m$^{-2}$ lower during foliar yellowing period and -3.59±3.90 W m$^{-2}$ lower during residue covering periods (Fig. 4e). $\alpha_{surf}$, $\beta_4$ and $Z_0$ have comparable impacts on LE during foliar yellowing and residue covering periods, with the changes of -1.46±1.69 W m$^{-2}$ and -2.17±2.98 W m$^{-2}$ induced by $\alpha_{surf}$ (ORC-A), and of -0.63±1.44 W m$^{-2}$ and -2.82±3.44 W m$^{-2}$ induced by $\beta_4$ and $Z_0$ (ORC-E), respectively. Different from H, the effect of crop residues on LE did not last beyond the residue covering period but ended in approximately 55 days within the residue covering period (Figs. S5b and S6b).

The corresponding changes in the sensible heat flux (H) are minor during the yellowing period, with an averaged decrease of -0.03±2.34 W m$^{-2}$ and moderate during the reside covering with averaged increase of 1.30±11.52 W m$^{-2}$, compared to the ORC-D simulation (Fig. 4f). Related to $\alpha_{surf}$ impacts (ORC-A), the surface-atmosphere energy exchange (ORC-E) in the two periods exerts a stronger influence on H with averaged increases of 3.46±1.04 W m$^{-2}$ and 15.44±13.61 W m$^{-2}$ during foliar yellowing and residue covering periods, respectively. The direct $\alpha_{surf}$ impacts cause an averaged increase of 0.14±2.22 W m$^{-2}$ and 1.41±10.09 W m$^{-2}$, respectively, during the same periods. Consistent with $T_{surf}$, the weakening influence of crop residues on H lasts 60 days after residue covering periods (Fig. S5c, Fig. S6c).

**3.4 Effect of soil residues on soil water availability on multi-year timescale**

The 10-year simulation for current climate shows a minor short-lived increase in SWC at the 12.5 % soil depth (relative root depth of winter wheat in ORCHIDEE-CROP model) due to presence of crop residues. Averaged over the simulation length and across 6 sites, total SWC is 0.19±0.27 kg m$^{-2}$ higher than in the simulation without crop residues (Fig. 5a). There was no statistically significant carry-over effect of SWC savings in one growing season to another at any site. This is because increased drainage losses compensated for reduced bare soil evaporation (Fig. S8). Soil temperature ($T_{soil}$) at 12.5 % soil depth penetrable by winter wheat roots decreased by an average of -0.19±0.44 °C over the 60 days of residue covering periods and across sites, and there is no significant cooling effect during the 60-day residue covering periods over 10 years (Fig. 5b). The largest reduction in $T_{soil}$ occurs after harvest coinciding with largest increase in $\alpha_{surf}$ due to residue presence compared to simulation without residue effects.





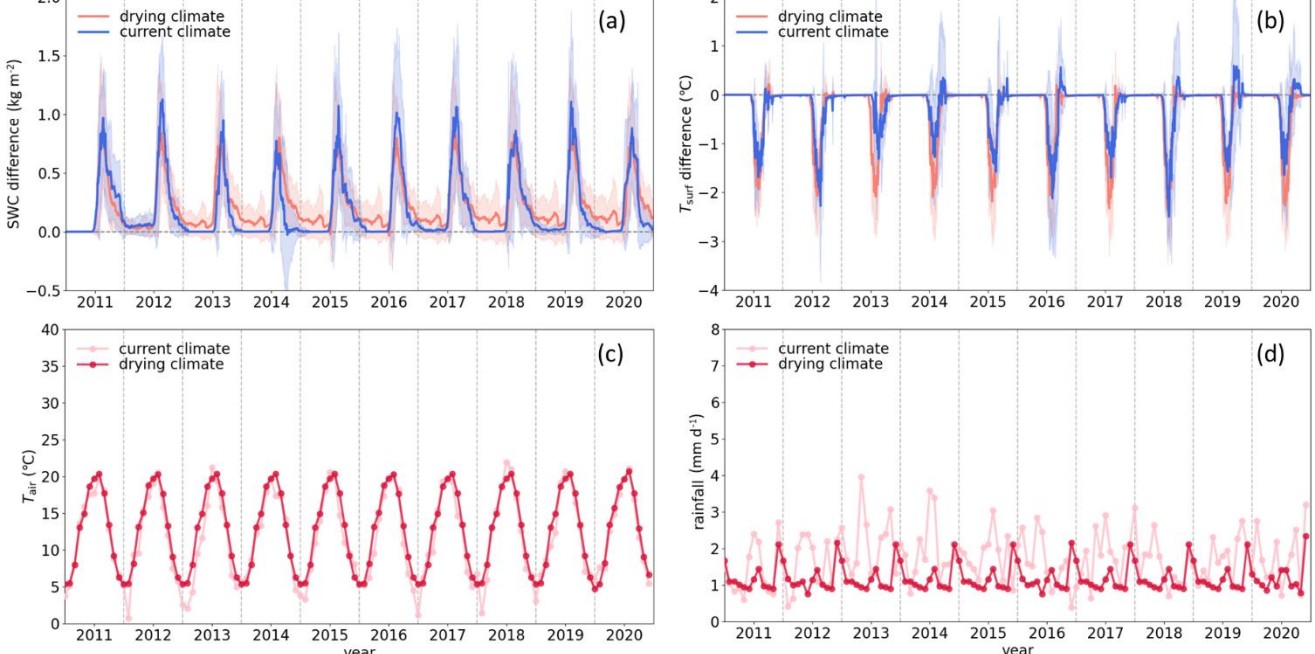

**Figure 5** The effect of new parameterization of surface albedo ($\alpha_{surf}$), surface roughness ($Z_0$) and soil conductance ($\beta_4$) on (a) soil water content (SWC) and (b) simulated daily soil temperature ($T_{soil}$) at 12.5 % soil depth under current and drying climate scenarios at 6 sites. Monthly temperature and rainfall are shown in (c) and (d), respectively. Shown are differences between results from ORC-AE and ORC-D. ORC-AE represents the effect of the modified $\alpha_{surf}$, $\beta_4$ and $Z_0$. ORC-D is the old model.

The 10-year simulation for drying climate shows a similar increase in SWC at 12.5 % soil depth due to the presence of crop residue with that for current climate, average over the simulation length and across sites of 0.22±0.21 kg m⁻² higher than in the simulation without crop residues (Fig. 5a). Compared with current scenario, the retention of SWC due to decreased $E_{soil}$ in drying scenario is comparable but exhibits no significant annual trend. However, it leaves more residual moisture available for the next growing season, potentially enhancing crop development by weakening water stress. This effect likely arises because water loss is governed primarily by evaporation in a drying scenario, as diminished water input (e.g., rainfall) (Fig. 5d) restricts both runoff and infiltration. Consequently, the $E_{soil}$ suppression by crop residues substantially increases soil water retention, underscoring their critical role in mitigating drought stress. We further found that SWC increase at 50 % soil depth has a statistically significant trend at all sites (not shown). This might be explained as, the absence of an annual trend at 12.5 % soil depth likely results from the rapid response of superficial SWC to short-term climate variability such as sporadic rainfall events, whereas the significant trend at 50 % soil depth reflects the cumulative effect of reduced $E_{soil}$ and enhanced deep-water retention over time. Average $T_{soil}$ at 12.5 % soil depth decreases with a statistically significant trend by





an average of -0.28±0.57 °C over years (Fig. 5b). Compared with current climate, the drying condition drives a larger magnitude of $T_{soil}$ decrease with the largest $\alpha_{surf}$ increase.

## 4. Discussion

### 4.1 Surface energy and land-atmosphere interaction jointly regulate surface temperature during foliar yellowing and residue covering periods

Foliar yellowing and residue coverage cool the surface across all sites by regulating the surface energy balance and redistributing net radiation to latent and sensible energy fluxes (Fig. 4a). The larger surface temperature ($T_{surf}$) decrease in ORC-A than that in ORC-E reveals that the cooling from the reduced net radiation ($R_n$) due to increased $\alpha_{surf}$ dominates the change in $T_{surf}$. This is in line with previous work, which indicated that no-tillage systems with residue coverage on European scale cooled the surface by about 2 °C during hot summer days due to increased $\alpha_{surf}$ (Davin et al., 2014).

While foliar yellowing and residue coverage induce an albedo-driven cooling effect across all 10 European sites, the interplay between surface energy available and the surface-atmosphere water-heat exchange causes substantial variation in the strength and periods of cooling. The increased surface roughness ($Z_0$) due to winter wheat and residue coverages in these two periods enhances surface-atmosphere turbulent heat exchange, potentially promoting more efficient heat dissipation and surface cooling (Winckler et al., 2019; Meier et al., 2022). For example, isolating the impact of increased $Z_0$ due to foliar yellowing and residue coverage (ORC-$Z_0$), both contribute more to the cooling impact than ORC-β across all sites (Fig. S9a). However, the extent to which this surface cooling persists also depends on the hydrological fluxes between surface and atmosphere. Crop residues potentially warm the surface by decreasing soil evaporation ($E_{soil}$) (Hu et al., 2018). It can be realized by lowering soil conductance ($\beta_4$) via shielding the soil from direct solar radiation and wind, and by reducing near-surface wind speed and enhancing aerodynamic drag via increasing $Z_0$ (Fig. 4a, Fig. S9c). Different from residue coverage, foliar yellowing has a minor influence on $E_{soil}$ due to the continuous winter wheat coverage on the soil (Fig. 4a). The associated decline in transpiration, driven by reduced stomatal conductance under lower net radiation, contributes to potential warming (not shown). Residue-induced suppression of $E_{soil}$ during residue coverage and $\alpha_{surf}$-induced decline in transpiration during foliar yellowing, both redirect energy partitioning from latent heat flux (LE) towards sensible heat flux (H), likely offsetting surface cooling (Ramos et al., 2024).

### 4.2 Distinct controls on LE and H during foliar yellowing and residue covering periods

To isolate the drivers of turbulent flux changes during crop yellowing and residue cover periods, we analyzed results from two key model configurations: the surface-atmosphere interaction experiments (ORC-E) and the radiative forcing experiments (ORC-A). Our simulations revealed that less surface energy available with a higher albedo during crop yellowing and the residues cover period, together with the subsequently altered turbulent fluxes, contributed to a weak





reduction in LE during these periods (Fig. 4e, Fig. S5b, Fig. S9d). In contrast, the surface-atmosphere interactions (ORC-E) dominate an increased H in both periods (Fig. 4f and Fig. S5c), and this effect was primarily mediated by $Z_0$ (Fig. S9b).

The different underlying drivers behind changes in LE and H can be explained by the following: LE represents the energy needed for water vaporization and depends on available energy and moisture transport efficiency (Chen et al., 2020). In contrast, H is driven by direct surface-atmosphere heat transfer via conduction and convection, governed by the land-air temperature gradient and turbulent heat exchange (Myhre et al., 2018). For example, we found that during the residue covering period, the change in H induced by effects of residues on $\beta_4$ and $Z_0$ (ORC-E) is much larger than by effects via surf

alone (ORC-A) (Fig. 4f). This is consistent with earlier findings pointing to a critical role of surface-atmosphere interactions in shaping energy partitioning and near-surface thermal dynamics (Koster et al., 2004; Seneviratne et al., 2010; Dare-Idowu et al., 2021; Denissen et al., 2022; Hsu et al., 2023). An opposite change in H during foliar yellowing and residue covering periods reflect the varying influence of land cover and surface properties on energy dynamics, driven by different ground covers (i.e. crops and residues) (McDermid et al., 2018).

The individual impacts of surf (ORC-A) and of surface-atmosphere exchange (ORC-E) on the changes in LE and H differ from the combined effect of both (ORC-AE), especially during the residue covering period (Fig. 4). This indicates a strong nonlinear interaction between energy and water dynamics driven by crop residues (Hsu et al., 2021). For example, the decreased LE reduces H during the residue covering period by weakening near-surface turbulence, leading to a significant negative impact on H not explained by ORC-A and ORC-E alone (Fig. 4f). The complex feedback mechanisms between

radiation partitioning and surface-atmosphere coupling (Hsu et al., 2021; Hsu et al., 2022) demands for modelling solutions which take into account both and their interactions.

### 4.3 Implication for climate change mitigation and adaptation

With approximately 60 % of global croplands having higher $T_{surf}$ than surrounding natural biome types, elevated temperatures, particularly during extreme heat and drought events, pose a major issue for crop yields and food security

(Lobell et al., 2012; Lesk et al., 2022; Chen et al., 2024). Our simulations across all sites suggest that residue coverage alleviates heat stress by decreasing maximum surface temperature ($T_{max}$) during approximately 20-day residue covering period and for up to 90 days afterward (Figs. S5e and S6e). This result aligns with previous findings on the cooling effect of crop residues (Davin et al., 2014; Webber et al., 2018; Su et al., 2022). Reduced surface $T_{max}$ after crop harvest mitigates heat stress on soil biota, providing a more favorable growing environment for subsequent crop cycles (Hatfield et al., 2015).

Sustained temperatures above 30 °C can stress mesophilic soil microbes, inhibiting decomposition and nutrient cycling (Pietikäinen et al., 2005). Our simulations show that residue cover reduces $T_{max}$ beyond 30 °C on 26 % of days during the residue period and 3.8 % of days afterward, underscoring its role in regulating soil thermal condition. This buffering effect modifies microbial activity and soil organic matter decomposition rates depending on specific climate and environment, thereby influencing greenhouse gas emissions from soil respiration (Knight et al., 2024).



The increase in H during residue covering period reveals an increased surface-air temperature gradient due to the decreased $T_{surf}$, possibly leading to a warmer lower atmosphere which cannot be quantified by the land surface model only (Fig. 4, Fig. S5c, Fig. S6c). This shift in energy partitioning from LE to H, driven by reduced $E_{soil}$ due to residue coverage, might mitigate surface warming but could intensify regional heat stress. Consequently, it introduces uncertainty regarding the role of crop residues as a climate mitigation strategy. Notably, the use of fixed air temperature forcing data partially explains the

increase in H because the constant air temperature ignores atmospheric feedback (Luyssaert et al., 2014). As a result, the climate mitigation potential of residue management remains uncertain, and its regional impact on energy partitioning cannot be fully determined. It is essential to consider not only surface cooling but also changes in air temperature, as the latter is more directly linked to climate dynamics. A coupled simulation, feasible with ORCHIDEE-CROP, would be needed for future studies.

Soil water content (SWC) dynamics play a key role in regulating the land-atmosphere interactions, as the increase in surface soil SWC, as due to residue effects on roughness and soil resistance, can decrease H partitioning via promoting $E_{soil}$ (Myhre et al., 2018). The 10-year simulations further highlight that residue coverage has the potential to increase SWC and thereby buffer drought stress, particularly under drier conditions with fewer rainfall events (Fig. 5, Fig. S8). In such condition, SWC remains high in the upper 1 m soil for extended periods of up to 3 months. An earlier review suggested that dry soil

condition might result in soil compaction and reduced hydraulic conductivity (Singh et al., 2018). It limits infiltration and slow downward water percolation, which might hinder the underlying mechanism of the SWC retention in the upper soil detected by ORCHIDEE-CROP model. Notably, soil texture modulates this effect. For example, rapid water infiltration into deeper layers in sandy soils with high drainage reduces surface SWC availability for cooling and plant uptake (Fatichi et al., 2020; Wankmüller et al., 2024), highlighting how soil hydraulics constrain residue efficacy. The sustained SWC also fosters

soil biota activity (e.g., fungi, earthworms), which enhance soil structure through aggregate formation and organic matter decomposition (Li et al., 2009). Improved soil structure increases water-holding capacity, creating a positive feedback loop where residue-induced SWC retention supports biological activity, which in turn amplifies long-term moisture retention. This synergy reveals that residue effects on SWC persist beyond immediate rainfall events, particularly in water-limited systems.

External factors such as tillage timing and precipitation frequency also affect the biophysical impacts of crop residues (Konapala et al., 2020). Early tillage, as observed at the FR-Gri site in 2018 (three weeks after harvest), leaves more SWC in the topsoil while removing the residue cover, temporarily boosting $E_{soil}$ and LE after the end of residue covering (Fig. S3). The strong soil water retention capacity of the silt loam soil at this site further contributed to this effect (Houot et al., 2000). This process underscores the importance of management practices in regulating the diverse hydrological and thermal effects

of residue covering. Targeted strategies are also essential to optimize water utilization, particularly facing warmer and drier climates (Swain et al., 2025).

## 4.4 Uncertainty and Limitations





Despite improvements in the ORCHIDEE-CROP model to simulate crop albedo changes from yellowing and residues and their biophysical climate impacts, there remain uncertainties and limitations.

First, the ORCHIDEE-CROP model has biases in simulating water and energy fluxes and the performance of the new model in capturing water-energy fluxes does not improve to the previous version when evaluated against site-level measurements at six sites (Fig. S10). Both the new and old models overestimate LE and H during foliar yellowing periods, while underestimating LE and overestimating H during residue covering periods. This overestimation during foliar yellowing periods might come from a systematically excessive vegetation–atmosphere coupling strength for crops in ORCHIDEE itself

compared with observation (Zhang et al., 2022). Observations indicate that LE is higher during periods of residue coverage compared to periods of bare soil (Fig. S12). However, it is not clear to what extent the observed difference in LE between periods is driven by differences in weather conditions or the presence/absence of residues. We manually adjusted the total conductance of soil evaporation, transpiration and interception in ORCHIDEE-CROP by scaling it with $1+f_{soil}$ during residue cover periods (Chapin et al., 2011). The simulations showed improved agreement with observed $T_{surf}$, LE and H

(Fig. S11), indicating that the net biophysical impact of residue cover is enhancing surface-atmosphere water and heat exchanges despite inhibiting soil evaporation, which is not well described in the current model. Dedicated field trials monitoring energy exchange would be required to validate the model but are not available to our knowledge.

Second, processes impacted by crop residues were not accounted for here such as water interception on the surface of residues, and uptake/release of water from residues. Third, the empirical parameterization of effects of crop pigmentation

and residue coverage on albedo neglects effects other than time, such as crop photosynthesis, soil moisture and residue density. Finally, the calibration of related parameters was based on a few scattered sites which might not be representative for larger scales.

## 5. Conclusion

Here we improved the simulation of winter wheat cultivar in the ORCHIDEE-CROP model by taking into account the

impact of foliar yellowing and crop residues on water and energy balance of winter wheat cultivations based on observations from 10 cropland sites. The simulated surface energy budget and partitioning of latent and sensible heat fluxes show a cooling impact of residues and crop yellowing. A complex interplay between radiative and turbulent processes controls the strength of these effects. The improved ORCHIDEE-CROP model, with its representation of land surface albedo, surface roughness, and soil conductance for residue covering, provides a valuable tool for quantifying biophysical impacts and

guiding localized residue management strategies at regional to global scale. By identifying optimal residue management practices, this approach can contribute to sustainable agriculture and inform policy development in the context of climate warming.

## Code and data availability

The model code used in this study is archived at https://doi.org/10.5281/zenodo.15230286 (Yu et al., 2025). All the data and the codes for data analysis and figure creation are available at https://doi.org/10.5281/zenodo.15234443 (Yu 2025). All data



used in this study are publicly available. The original ecosystem flux data at available sites can be derived from the Integrated Carbon Observation System (ICOS) Data Portal at https://data.icos-cp.eu/. The MODIS LAI data (MCD15A3H) can be downloaded from Land Processes Distributed Active Archive Center (https://lpdaac.usgs.gov/products/mcd15a3hv006/ DOI: https://doi.org/10.5067/MODIS/MCD15A3H.006). The MODIS aerosol product (MO(Y)D04_L2) can be downloaded from Level-1 and Atmosphere Archive & Distribution System Distributed Active Archive Center (https://ladsweb.modaps.eosdis.nasa.gov/ DOI: https://doi.org/10.5067/MODIS/MOD04_L2.061). The Sentinel-2 bare soil albedo datasets can be downloaded from https://doi.org/10.5281/zenodo.15271053 (Yu 2025).

## Author contributions

DG, RL, and PC conceptualized and designed the study. KY modified and implemented the model, conducted formal analysis of the results, and led the writing of the original draft with contributions from all co-authors. AP and MC contributed data curation by providing management information from ClieNFarm sites. All authors participated in reviewing, editing, and finalizing the manuscript.

## Competing interests

The contact author has declared that none of the authors has any competing interests.

## Acknowledgements

The authors gratefully acknowledge the ClieNFarm project partners for providing spatial datasets (shapefiles) and detailed management practice records for winter wheat fields within their agricultural sites. We also extend our thanks to the ORCHIDEE model development team for their technical support and expertise during the model implementation process.

## Financial support

This project is funded by ClieNFarms project (Grant Agreement ID: 101036822).



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
