# Peer review of "Resolving effects of leaf pigmentation changes and plant residue on the energy balance of winter wheat cultivation in the ORCHIDEE-CROP model"

_EGUsphere, 2025_

## Author Comment (AC1)

**Response to Reviewer 2**

**General comments**

Yu et al present the ORCHIDEE-CROP model that incorporates a time-varying surface albedo that considers foliar yellowing and crop residue. The effect of crop residue on surface roughness, surface temperature, and surface energy partitioning was also analyzed. Overall, the work is solid and addresses effects not considered in current models, while it may benefit from some clarification on methods and interpretations. I suggest publication of the manuscript after minor revision.

[Response] Thank you so much for your time in reviewing our manuscript and for providing your constructive comments and suggestions. The valuable comments helped us improve the paper. Following the comments, our main revision is as below:

(1) Clarified the magnitude of impacts of foliar yellowing and residue covering on water and heat variables in the Abstract, and included the relative changes in targeted variables between the improved model and the initial version in the results sections; (2) Expanded the discussion of the uncertainty and spatiotemporal variability of residue impacts in sections 4.2 and 4.5; (3) Supplemented more details about the calibration of crop development and harvest timing in sections 2.2.5 and 2.3.

Please find our detailed responses to all of your comments below.

**Major Comments**

1. In several places across the manuscript (e.g., L36), the manuscript claims a significant impact on surface energy balance. However, in most scenarios, the impact seems small (less than a few W/m2) to me, and the uncertainty is at similar magnitudes as the difference. I think the magnitude of the differences can be better addressed.

[Response] We agree that the absolute changes in surface energy fluxes (i.e., LE and H) are generally modest (Fig. 4e and f) and at similar magnitudes as the uncertainty. However, these small values of the mean of differences across the sites mask substantial spatiotemporal variability, especially in residue effects. During the foliar yellowing period, the ~20-day increase in surface albedo (~0.02) only slightly perturbs the surface energy balance. At this stage, the balance between latent and sensible heat fluxes is primarily controlled by crop transpiration when the soil is invisible. Because the available surface energy remains nearly unchanged, total flux variations remain minor.

During the residue covering period, flux responses exhibit pronounced spatiotemporal heterogeneity with a relatively large standard deviation. Temporally, residue-induced albedo enhancement evolves with time. It peaks within the first 15 days and weakens as residues decompose (Figs. 3 and S4), producing short-lived yet locally significant perturbations in water and heat processes which lasts about 5-30 days after the peak and also persist for 1-2 months beyond residue removal through tillage or natural decomposition (Fig. S5 and S6). Spatially, variations in climate, soil properties and management practices amplify local differences in residue impacts. For example, we found that during the residue covering period, BE-Lon exhibits the largest mean relative decrease in LE in the improved model compared to the initial version (-27.69±11.62%), whereas the change at DE-Geb is minimal (-6.10±33.48%). This contrast arises because the higher soil moisture and sandy texture at BE-Lon support greater evaporative fluxes, whereas the limited soil water and clay-rich soil at DE-Geb favor energy dissipation as sensible rather than latent heat (Dumont et al., 2023; Buysse et al., 2023). Moreover, even with fixed 50% decrease of soil conductance and residue height parameters of 0.5 m (see sections 2.3.3 and 2.3.4), the model reproduces clear site-to-site variability in the magnitude and direction of residue effects on water and heat fluxes.

In addition, long-term simulations show the accumulated residue impact on soil moisture and soil temperature. The 10-year simulation under the drying climate scenario reveals a statistically significant cooling trend, despite a mean annual temperature decrease of only 0.3 °C (Fig. 5). This demonstrates that even modest instantaneous flux changes can produce meaningful long-term surface cooling effects.

Therefore, evaluating residue effects requires a context-specific approach that accounts for the interactions among climate, soil, and crop characteristics. Process-based land surface models provide an effective framework for such assessments at varying scales, as they explicitly represent the coupled biophysical and ecological mechanisms controlling surface energy and water exchanges. This approach provides an opportunity to guide residue management strategies under variable environmental settings, emphasizing the importance of this work.

We revised the manuscript to more accurately describe the strength of residue effects in the Abstract.

Lines 36-37: "This study underscores that crop pigmentation has a minor influence on heat and water budgets, while residues moderately modulate surface energy partitioning with significant spatial heterogeneity, and demonstrates the potential of its management for climate mitigation."

We also expanded the discussions in sections 4.2 and 4.5 with the following sentences:

Lines 564-578: "It should be noted that the averaged difference in surface energy fluxes (i.e., LE and H) over all sites between the new and old models are generally modest (Fig. 4 (e) and (f)) and at similar magnitudes as model uncertainty, which masks the substantial spatiotemporal variability in residue effects during the residue covering period. Temporally, residue-induced albedo enhancement peaks within the first 15 days after harvest and weakens as residues decompose, causing short-lived but locally significant perturbations in water and heat fluxes (Fig. 3, Fig. S4-S5). The residue impact on the water-heat processes persists for an additional 1-2 months beyond residue removal through tillage or natural decomposition (Fig. S6). Spatial variations in climate, soil properties and management practices amplify local differences in residue impacts. For example, we found that during the residue covering period, BE-Lon exhibits the largest mean relative decrease in LE in the improved model compared to the initial version (-27.69±11.62%), whereas the change at DE-Geb is minimal (-6.10±33.48%). This contrast arises because the higher soil moisture and sandy texture at BE-Lon support greater evaporative fluxes, whereas the limited soil water and clay-rich soil at DE-Geb favor energy dissipation as sensible rather than latent heat (Dumont et al., 2023; Buysse et al., 2023). In addition, long-term simulations show the accumulated residue impact on heat budgets. The 10-year simulation under the drying climate scenario shows a low but statistically significant cooling trend, with a yearly average of -0.3 °C (Fig. 5). This demonstrates that even modest instantaneous flux changes can generate meaningful long-term surface cooling effects."

Lines 674-678: "[...], evaluating residue effects requires a context-specific approach that accounts for the interactions among climate, soil, and crop characteristics. Process-based land surface models provide an effective framework for such assessments at varying scales, as they explicitly represent the coupled biophysical and ecological mechanisms controlling surface energy and water exchanges. This approach provides an opportunity to guide residue management strategies under variable environmental settings. [...]."

2. L231, L242: It would be good to explain how the calibrations were performed in the supplementary. [Response] We appreciate this suggestion for improving this analysis. For the calibration of crop development, it has been comprehensively described and published in Su et al. (2025, https://doi.org/10.1016/j.eja.2025.127723). To avoid unnecessary repetition, we did not reproduce all

methodological details here. Instead, we now include a concise description of the calibration procedure in the revised manuscript with the reference of this paper:

Line 252-258: "The maturity and harvesting dates were decided by the growing degree days and grain water content in the crop development processes of the ORCHIDEE-CROP model. The threshold of meeting harvesting conditions was calibrated following the procedure described by Su et al. (2025). Parameters governing the growing degree day requirements for key phenological stages were optimized using observed phenological records (planting, flowering, and harvest dates) from 2890 DWD climate stations across Germany (949 sites for maize and 1941 for wheat; https://opendata.dwd.de/climate\_environment/). The optimal parameter sets for wheat and maize were identified by minimizing the root mean square error (RMSE) between simulated and observed harvest dates across all sites."

For the calibration of the duration between maturity and harvest, we run this model iteratively with a range of potential durations between maturity and harvest (from 5 to 40 days at 2-day increments). The duration that resulted in a simulated harvest date perfectly matching the observed harvest date was then selected for each site year. This approach ensures that the simulated phenology aligns closely with site-specific agricultural practices and environmental conditions.

We included this duration information in Supplementary Table 1, and supplemented the details in section 2.3 with the following sentences:

Line 265-270: "In addition, to align the total crop development period (from sowing to harvest) between simulations and observations at each site, the duration between maturity and harvesting in ORCHIDEE-CROP was calibrated using recorded harvesting dates at each site. In each winter wheat year at each site, simulations were performed iteratively with potential maturity-harvest intervals ranging from 5 to 40 days at 2-day increments. The optimal duration for each case was identified when the simulated harvest date matched the recorded management date. The calibrated durations are summarized in Table S1."

- 3. L312: The equation number seems incorrect
  [Response] We replaced the equation numbers from Eqs.15 and 19 to Eqs. 12 and 14 (Line 341).
  - 4. L355: How was the asurf model trained? Random forest is mentioned here, but Section 2.4.2 describes a direct fitting of the parameters in Eqs. 13 and 14.

[Response] Thank you for pointing us to this writing mistake. The  $\alpha_{surf}$  training is based on polynomial regression rather than a random forest model based on data from 10 sites with 33 winter wheat site-years. We used a linear incremental function (Eq. 12) to fit  $\alpha_{surf}$  increase during the foliar yellowing and the first 15-day residue covering periods, and an exponentially decreasing function (Eq. 14) to describe the  $\alpha_{surf}$  decrease during the residue covering period (sections 2.3.1, 2.3.2 and 2.4). In addition, we corrected the text in section 3.1, where we referred to this procedure, by adding the following sentences:

Line 390-392: "The least squares regression in these two fitting models captures the mean tendency of the data but fails to reproduce extreme variations, as it minimizes average errors rather than explaining outliers."

5. Figure 4: It would be good to show the relative difference in the main text or supplementary to provide more context.

[Response] We added the relative difference of  $E_{soil}$ ,  $T_{surf}$ , LE and H between the new and old models in section 3.2-3.4, Please see the revised manuscript.

Here we give a modified example (**Lines 420-426**): "ORC-AE simulates slightly lower soil evaporation ( $E_{soil}$ ) during foliar yellowing and residue covering periods than the standard version of ORCHIDEE-CROP (ORC-D), with averaged decreases of -0.01±0.04 mm d-1 (-3.07±13.00%) and -0.11±0.14 mm d-1 (-13.34±32.60%), respectively (Fig. 4a). The effects of soil conductance ( $\beta_4$ ) and soil roughness( $Z_0$ ) reduced  $E_{soil}$  on average by -0.01±0.03 mm/d (-2.86±9.91%) and -0.09±0.12 mm d-1 (-9.69±38.16%) (ORC-E) during foliar yellowing and residue covering periods, respectively. Impacts on  $\alpha_{surf}$  alone cause reductions of respectively -0.01±0.04 mm d-1 (-2.30±15.50%) and -0.06±0.10 mm d-1 (-9.59±22.43%) (ORC-A) for the same periods. The influence of crop residues on  $E_{soil}$  disappears within 40 days after residue coverage (Fig. S5a, Fig. S6a)."

**Reference**

Buysse, P., Depuydt, J., and Loubet, B.: ETC L2 ARCHIVE, Grignon, 2022-12-31–2023-09-30, ICOS RI, doi:10.18160/jU9ftXkRFqo-dc91cDfOYuA8, 2023.

Dumont, B., Bogaerts, G., Chopin, H., De Ligne, A., Demoulin, L., Faurès, A., Heinesch, B., Longdoz, B., Manise, T., and Orgun, A.: ETC L2 ARCHIVE, Lonzee, 2022-12-31-2023-09-30, ICOS RI, https://doi.org/10.18160/JmEjmkD1hSr4YIuo5Uu-L69a, 2023.

Su, Y., Lauerwald, R., Makowski, D., Viovy, N., Guilpart, N., Zhu, P., Gabrielle, B., & Ciais, P. Future warming increases the chance of success of maize-wheat double cropping in Europe. European Journal of Agronomy, 170, 127723. https://doi.org/10.1016/j.eja.2025.127723, 2025.

---

## Author Comment (AC2)

**Response to Reviewer 1**

**General comments**

This manuscript investigates the effects of leaf pigmentation changes and crop residues on the surface energy balance of winter wheat, by improving the ORCHIDEE-CROP model with dynamic albedo, soil evaporation, and surface roughness parameterizations. The study is clearly presented, methodologically sound, and provides valuable insights into biophysical processes often overlooked in land surface models. Some aspects, however, would benefit from refinement:

[Response] Thank you so much for your time in reviewing our manuscript and for your positive feedback on our work. Following your suggestions, the revision of this manuscript is as follows:

(1) Added a sensitivity analysis of key model parameters to analyze the variability of residue effects. The corresponding methodology and results of the sensitivity test were described in section 2.6 and section 3.5; (2) Expanded the discussion of model limitations and uncertainties in section 4.5.

For more details, please find our responses to all of your comments below.

**Major Comments**

1. The site-to-site variability in residue effects (on evaporation and roughness) deserves more explicit discussion of limitations.

[Response] We agree about the need to clarify site-to-site variability in residue effects. We acknowledge that in reality, residue characteristics and environmental conditions differ substantially among sites. Variations in residue amount, texture, and decomposition rate, as well as differences in soil properties, humidity, wind, and radiation, all influence soil evaporation and surface roughness. These factors together lead to diverse residue effects on energy and water fluxes across locations. In the current version of ORCHIDEE-CROP, however, the representation of residue effects on soil evaporation and surface roughness is intentionally simplified to balance model complexity with the scarcity of data to constrain parameterization. Specifically, the impact of residues on soil conductance ( $\beta_4$ ) was represented by an initial reduction factor of 0.5 at the start of the residue covering period that linearly decreases with residue decomposition (Section 2.3.3). While residue influence on surface roughness ( $Z_0$ ) was prescribed using a fixed residue height (0.5 m), derived from average measurements across five winter-wheat sites (Section 2.3.4, Table S3). These assumptions are uniform across all sites and therefore cannot explicitly represent local variation in residue characteristics, soil properties, or atmospheric conditions.

Despite these simplifications, site-to-site variability in simulated residue effects still emerges from interactions between these uniform parameters and local environmental conditions. Differences in climate (temperature, humidity, radiation, wind speed) and soil state (texture, moisture) modulate the realized impact of residues on soil evaporation and energy partitioning, leading to variable outcomes across sites (Figs. 4, S5-S6, and S9).

To clarify these points in the manuscript, first, we added text in section 4.5 explicitly acknowledging the uniform treatment and its implications with the following sentences:

Lines 634-643: "One possible source of bias during the residue covering period may be the simplification of model parameters. In the present version of ORCHIDEE-CROP, the effects of crop residues on surface-atmosphere coupling are quantified by modulating  $\beta_4$  and  $Z_0$ . Both variables are described with uniform assumptions to balance model complexity with the scarcity of data to constrain parameterization. Specifically, the impact of residues on  $\beta_4$  was represented by an initial reduction

factor of 0.5 at the start of the residue covering (Section 2.3.3). While residue influence on  $Z_0$  was prescribed using a fixed residue height (0.5 m), derived from the average of measurements across five winter-wheat sites (Section 2.3.4). These parameter choices are uniform across all sites and therefore cannot explicitly represent local variation in residue characteristics, soil properties, atmospheric conditions and management practices. As a result, the model is incapable of fully resolving site-specific residue impacts, which potentially contribute to the bias of simulated LE and H at certain sites."

Second, we performed a sensitivity test showing how varying the parameters  $\beta_4$  and residue height modifies results and uncertainty ranges (see details in Comment 2). The new sensitivity analysis indicates that  $Z_0$ , determined by residue height, and  $\beta_4$  exert primary controls on the surface-atmosphere water-heat exchange (Fig. S13). A  $\pm 30\%$  change in residue height consistently weakens the amplitude of flux responses, suggesting that the baseline value of 0.5 m setting may overestimate residue effects on evaporation and turbulent fluxes. Variations in  $\beta_4$  strongly influence surface temperature, soil evaporation, sensible and latent heat fluxes, underscoring that the site-specific parametrisation of  $\beta_4$  is essential to capture site-to-site differences in the effects of crop residues. The strong sensitivity to  $\beta_4$  also highlights the need to better constrain this parameter in future model developments with an appropriate spatial dataset to force the model. Finally, we expanded the discussion to outline future model improvements that could better capture local variability through site-specific residue parameterization, explicit residue energy budgets, and linkage to management on residue cover.

Line 678-683: "[...]. Future model developments should include explicit residue modules and site-specific parameterization to better capture spatial heterogeneity in residue impacts on energy and water fluxes. For example, integrating the canopy interception modules developed by crop models to ORCHIDEE-CROP is a good strategy to better represent the residue impact on the hydrological dynamics, such as modules from CropSyst and RZWQM (Kozak et al., 2007). Moreover, an open-sourced global database from dedicated field trials monitoring energy exchange is required for parametrizing and evaluating this model development."

2. The treatment of uncertainties and sensitivity to input data and climate scenarios could be expanded.

[Response] We agree with this valuable comment regarding the treatment of uncertainties and the sensitivity of model results to input data and climate scenarios. In the revised manuscript, we have expanded both the uncertainty discussion and the sensitivity analyses to address this point more explicitly.

(1) To assess the influence of climate variability on residue impacts, we performed 10-year simulations under both current and drying climate scenarios (section 3.4). Under current climate conditions, residues slightly increased soil water content at 12.5 % soil depth by 0.19±0.27 kg m-2, with no significant multi-year carry-over due to compensating drainage losses. In the drying scenario, the soil water increase (0.22±0.21 kg m-2) was comparable in magnitude but more effective in retaining moisture between seasons, reflecting the dominant control of evaporation under reduced precipitation. The larger and statistically significant soil temperature decrease (-0.28±0.57 °C) under drying conditions further highlights enhanced surface cooling linked to increased surface albedo. Together, these experiments demonstrate that the magnitude and persistence of residue impacts on soil water content, soil temperature, and energy partitioning are climate-dependent, and the model captures interactions between residue properties and hydrometeorological conditions under specific climate conditions.

- (2) The uncertainty induced by input data mainly comes from three aspects. First, while site-based meteorological forcing (e.g., air temperature, wind speed, humidity) ensures site representativeness, 2 ClieNFarm sites lack flux observations and were therefore driven by the 6-hourly 0.5° CRU-JRA (v2.4) product. This substitution inevitably limits the ability of this model to reproduce site-specific water and energy dynamics. Second, the bare soil albedo dataset used to estimate surface albedo carries uncertainties related to quality-restricted training samples (Yu et al., in review). For example, soil samples cannot be extracted under cloudy, snow or crop covering conditions. The vegetation index thresholding used to identify bare soil may include mixed surfaces, such as fallow with crop residues, introducing bias in soil albedo retrievals. Also, the spatial aggregation from 300 m to 0.5° smoothes the field variability of bare soil albedo, reducing the reliability of the model simulation. Third, the incomplete management information in this model, including chemical applications such as fertilizers or physical operations like tillage, constrains the ability of this model to reproduce observed variations in crop phenology, soil water content, and surface fluxes across years.
- (3) We additionally performed a parameter sensitivity analysis to test model uncertainty associated with key residue-related parameters in the improved ORCHIDEE-CROP version (section 2.6, section 3.5, section 4.5 and Figure S13). The analysis tested how surface temperature ( $T_{surf}$ ), soil evaporation ( $E_{soil}$ ), sensible heat flux (H), and latent heat flux (LE) respond to  $\pm 10\%$ ,  $\pm 20\%$ , and  $\pm 30\%$  perturbations of individual parameters, including residue height (Height) related with surface roughness, soil conductance ( $\beta_4$ ) regulating soil evaporation, duration of surface albedo increase during residue covering period ( $D_{up}$ ), empirical slopes ( $k_1$  and  $k_4$ ) in  $D_{up}$  and in surface albedo decrease stages in the modelling.

Sensitivity analyses demonstrate distinct and nonlinear sensitivities of surface energy and water fluxes to the tested parameters. Specifically, the sensitivity of  $T_{surf}$  is strongest to  $k_1$ , which controls the rate and envelope of 15-day albedo increase and thus regulates net surface radiance. While the response of  $T_{surf}$  to  $\beta_4$  is minor ( $\leq 8\%$ ), indicating that radiative, rather than hydrological processes, dominate  $T_{surf}$  variability. The small sensitivity to  $D_{up}$  implies that the rate of surface albedo change, rather than the duration, dominantly contributes to the  $T_{surf}$ . It is understandable that  $E_{soil}$  shows the strongest dependence on  $\beta_4$ , with  $\pm 30\%$  perturbations resulting in 62-101% changes in  $E_{soil}$ . Parameters  $k_1$ ,  $k_4$ , and  $D_{up}$  have secondary effects (changes of  $\sim 18-36\%$ ). These three parameters shape the availability of surface energy for  $E_{soil}$ . It indicates that the  $E_{soil}$  is jointly controlled by albedo-induced surface radiation and evaporation ability, which aligns with our previous analysis (Figure 4(a)). The H and LE exhibit sensitivity to  $D_{up}$ ,  $k_1$ , and  $\beta_4$ , consistent with the strong coupling between surface energy partitioning and atmospheric forcing (Figure 4, Figure S5, S6 and S9). We note that the strong sensitivity of  $E_{soil}$  to energy changes compared to the lagged energy balance component LE.

We included the methodology of sensitivity analysis in section 2.6, and the result in section 3.5 with the following sentences (Figure S13 is added in the supplementary information):

- (1) Lines 369-372: "Section 2.6 sensitivity analysis: We performed sensitivity tests of the major parameters (i.e.  $k_1$ ,  $k_4$ ,  $\beta_4$ , Height, duration of  $\alpha_{surf}$  increase during residue covering period  $(D_{up})$ ) linking to the  $\alpha_{surf}$  variation and the responses of the energy and water budgets, particularly for the ones without enough constraints from field observations. The sensitivity test was conducted by changing the parameters by  $\pm 10\%$ ,  $\pm 20\%$  and  $\pm 30\%$  from the initial calibrated values. Their impacts on  $T_{surf}$ , H,  $E_{soil}$  and LE were evaluated."
- (2) Lines 501-516: "Section 3.5 Sensitivity of energy and water processes to parameters: Sensitivity analyses reveal pronounced nonlinear responses of land surface water and energy processes to

parameter dynamics (Fig. S13). For  $T_{surf}$ ,  $k_I$  exhibits the highest sensitivity, as it governs the rate and ceiling of 15-day surface albedo increase and consequently modulates available surface radiation during the residue covering period (Fig. S13a). A  $\pm 30\%$  perturbation in  $k_I$  induces 8.69% and 30.78% enhancements in surface cooling, respectively. In contrast,  $T_{surf}$  is only weakly affected by the hydrological parameter  $\beta_4$  (-4.46% / +2.98% change in surface cooling for  $\pm 30\%$  variations), consistent with the scenario simulations (Fig. 4b). The parameter  $D_{up}$  also shows limited influence on  $T_{surf}$  (-8.35% / +2.53% change in cooling for  $\pm 30\%$  variations).

For  $E_{soil}$ ,  $\beta_4$  is the dominant control (Fig. S13b). A decrease or increase in  $\beta_4$  from the baseline by 30% results in a 62.48% reduction or 100.85% increase in the decline of  $E_{soil}$ , respectively. The parameters  $k_1$ ,  $k_4$ , and  $D_{up}$  also influence  $E_{soil}$ , with a  $\pm 30\%$  variation leading to -32.40% / -35.53%, -17.72% / +22.35%, and -29.49% / -18.57% changes, respectively.

For H,  $D_{up}$  exerts the strongest control, followed by  $k_I$  and  $\beta_4$  (Fig. S13c). Adjusting these parameters by  $\pm 30\%$  alters H by -10.53% / +56.34%, -42.55% / +36.77%, and -27.60% / +27.33%, respectively.  $\beta_4$  is also the most sensitive parameter for LE, reflecting its dependence on  $E_{soil}$ .  $\pm 30\%$  variations in  $\beta_4$  yield +89.78% / -62.16% changes in LE (Fig. S13d). In contrast, the strong sensitivity of  $E_{soil}$  to  $k_I$  does not propagate to LE, with only -2.66% / +8.21% changes observed under equivalent  $k_I$  perturbations (Fig. S13c and d)."

Figure S13 Relative change in residue impacts on surface temperature ( $T_{surf}$ , (a)), soil evaporation ( $E_{soil}$ , (b)), sensible heat flux (H, (c)) and latent heat flux (LE, (d)) by varying  $\pm 10$ ,  $\pm 20$  and  $\pm 30\%$  of residue height (Height), soil conductance ( $\beta_4$ ), duration of surface albedo increase during residue covering period ( $D_{up}$ ), slopes ( $k_I$  and  $k_4$ ) in  $D_{up}$  and in surface albedo decrease stages in the modelling. Residue impacts are quantified by relative changes of variables above between ORC-AE and ORC-D. ORC-AE adjusts surface albedo,  $\beta_4$  and surface roughness together in the ORCHIDEE-CROP model. ORC-D is the initial model version. Parameters are changed one by one, while the others are kept the same. '(+)' and '(-)' behind the subplot titles represent the increase and decrease of variables caused by parameters at baseline condition.

Finally, we revised and expanded the discussion of uncertainty sources in section 4.5 to clarify the limitations and uncertainty of this model improvement:

Lines 644-647: "[...], The sensitivity analysis also highlights the uncertainties introduced by parameter selection (Fig. S13). The  $\beta_4$  and  $k_1$  exert strong control over the spatiotemporal partitioning of available surface energy between LE and H. The use of fixed parameter values and limited calibration at a few sites inevitably contributes to model uncertainty and constrains the representation of local variability.

Lines 651-673: "[...], A test of quantifying the residue impact on  $E_{soil}$  at 15 field experiments distributed globally shows that residue cover exceeding 80% leads to an approximately 20-30% reduction in  $E_{soil}$  during the residue covering period (not shown). Therefore, the assumed 50% initial decline in  $\beta_4$  in the improved model (Eq.16) might overestimate residue impacts, potentially explaining the lower  $E_{soil}$  and LE simulated across sites in the new model compared with both the initial model version and observations (Fig. S10).

In addition, specific hydrological processes impacted by crop residues were not accounted for here, such as water interception on the surface of residues, and uptake/release of water from residues (Kozak et al., 2007; Swella et al., 2015). Rainfall interception by residues alters the timing and magnitude of soil water input and enhances evaporation from residue surfaces, thereby modifying surface moisture and heat exchanges (Mitchell et al., 2012; Thapa et al., 2021). However, contrasting evaporation drivers in soil vs. residue layers introduce complexity to surface evaporation processes. The omission of these processes likely increases model uncertainty in biophysical simulations under variable rainfall conditions.

The input data used to drive the model introduces several structural uncertainties. First, while site-based meteorological forcing (e.g., air temperature, wind speed, humidity) ensures high representativeness, at two ClieNFarm sites meteorological observations were missing and simulations were instead driven by the 6-hourly 0.5° CRU-JRA (v2.4) product. This substitution inevitably limits the ability of our model to reproduce site-specific water and energy dynamics.

Second, the bare soil albedo dataset used to estimate surface albedo carries uncertainties related to quality-restricted training samples (Yu et al., in review). For example, soil samples cannot be extracted under cloud, snow or crop covering conditions. The vegetation index thresholding used to identify bare soil may also include mixed surfaces such as crop residues, introducing bias in soil albedo retrievals. Also, the spatial aggregation from 300 m to 0.5° smoothes the field variability of bare soil albedo, reducing the reliability of the model simulation.

Third, the incomplete representation of management practices in our model, including fertilizer and pesticide application as well as physical operations like tillage, limits the model's capacity to reproduce observed variations in crop phenology, soil water content, and surface fluxes across years."

3. Figures are informative but some captions and explanations are too brief; adding interpretive detail would improve readability.

[Response] We apologize for the unclear information about our figures and thank you for helping us to improve it. We included more detailed information in Figure 1-5 and Supplementary figure 3-12:

**In the main text:**

(1) Lines 101-119 in Figure 1: 'Figure 1 The procedure of parameterization of crop and residue albedo  $(\alpha_{surf}$  and  $\alpha_{res})$ , soil evaporation  $(E_{soil})$  and surface roughness  $(Z_0)$  in the ORCHIDEE-CROP model. Panel

- (A) illustrates the input datasets for albedo calibration.  $\alpha_{soil}$  and  $\alpha_{surf}$  are bare soil albedo ( $\alpha_{soil}$ ) and surface albedo ( $\alpha_{surf}$ ), respectively.  $f_{crop}$  and  $f_{res}$  are the gridded fractions of crop ( $f_{crop}$ ) and residues ( $f_{res}$ ), respectively. SW IN and SW OUT are half-hourly incoming (SW IN) and outgoing (SW OUT) solar radiance observed at 7 eddy-covariance (EC) sites. Panel (B) shows the identification of foliar yellowing and residue covering periods based on the time series of  $\alpha_{surf}$  (black curve) and leaf area index (LAI, green curve).  $T_{0.5}$  represent the dates of maximum LAI ( $T_0$ , green dotted vertical line), albedo increase start ( $T_1$ , orange dotted vertical line), harvest ( $T_2$ , grey solid vertical line) and albedo increase end ( $T_3$ , shallow-blue dotted vertical line), 30 days after harvest ( $T_4$ , grey solid vertical line) and tillage (equals to 90 if no tillage) ( $T_5$ , dark-blue dotted vertical line).  $\alpha_{min}$  and  $\alpha_{max}$  are the minimum and maximum surface albedo on  $T_l$  and  $T_3$ . The shallow-green, orange, yellow and brown areas show the conceptual growing, maturity, residue covering and bare soil periods of winter wheat. Panel (C) describes trend fittings for crop albedo ( $\alpha_{crop}$ ) and  $\alpha_{res}$ . Features in two plots have the same meanings as those in Panel (B).  $f_{crop, yellowing}$  is the fraction of crop during the foliar yellowing period. RC is the duration of  $T_2$ and  $T_5$ ,  $k_1$ ,  $k_2$ ,  $k_3$  and  $k_4$  are the fitting parameters during foliar yellowing and residue covering periods. day and t are the number of days since  $T_1$  and  $T_2$ , respectively. Panel (D) illustrates parameterization of  $\alpha_{surf}$  as a weighted combination of  $\alpha_{crop}$ ,  $\alpha_{res}$  and  $\alpha_{soil}$  in the ORCHIDEE-CROP model. The daily  $\alpha_{surf}$  in orange in the upper plot is derived from the new calibrated process based on  $k_l$ - $k_d$ , compared to the old model (plot below). Panel (E) presents parameterization of surface-atmosphere exchange.  $\beta$  is the resistance coefficient;  $r_{au}$ , speed,  $C_d$ ,  $q_{surf}$  and  $q_{air}$  are the air density  $(r_{au})$ , wind speed (speed), drag coefficient ( $\mathcal{C}_d$ ), saturated surface air moisture ( $q_{surf}$ ) and specific humidity ( $q_{air}$ ), respectively.
- (2) Lines 394-397 in Figure 2: 'Figure 2 The comparison of daily surface albedo ( $\alpha_{surf}$ ) predictions based on the new parameterization (Figure 1) with independent observation from five sites in Europe during (a) the foliar yellowing and (b) residue covering periods, with coefficient of determination ( $R^2$ ) and root mean square error (RMSE) in the bottom-right corners. The dotted black line is the 1:1 line.'
- (3) Lines 410-416 in Figure 3: 'Figure 3 The comparison of surface albedo ( $\alpha_{surf}$ ) predicted from the old (orange dots, ORC-AE) and new (red dots, ORC-D) ORCHIDEE-CROP models and observations at Grignon site in France (gray dots) in 2018. ORC-AE represents the new model version with effects of the modified  $\alpha_{surf}$  and the refined soil conductance ( $\beta_4$ ) and surface roughness ( $Z_0$ ), while ORC-D suggests the initial version of the model. The observed  $\alpha_{surf}$  is computed from site radiation measurements through the Integrated Carbon Observation System (ICOS) Data Portal. The green, black and blue solid lines are the simulated start of the foliar yellowing period (shallow green area) ( $T_{yellowing, start}$ ), harvesting date ( $T_{harvest}$ ) and the end of residue covering period (shallow blue area) ( $T_{res, end}$ ), respectively.'
- (4) Lines 428-436 in Figure 4: We included: "[...], the grey dotted vertical line in each plot represents zero on the y-axis." at the end of the Figure 4 caption.
- (5) Lines 479-484 in Figure 5: 'Figure 5 The 10-year cumulated effect of new parameterization of surface albedo ( $\alpha_{surf}$ ), surface roughness ( $Z_0$ ) and soil conductance ( $\beta_4$ ) on (a) soil water content (SWC) and (b) simulated daily soil temperature ( $T_{soil}$ ) at 12.5 % soil depth under current and drying climate scenarios at 6 sites from 2011 to 2020. Monthly temperature and rainfall are obtained from the 6-hourly meteorological variables of the 0.5° CRU-JRA product (v2.4), shown in (c) and (d). Shown are the differences between the results from ORC-AE and ORC-D. ORC-AE represents the new model version with effects of the modified  $\alpha_{surf}$  and the refined  $\beta_4$  and  $Z_0$ , while ORC-D suggests the initial version of the model.'

**In the supplementary information:**

- (1) Lines 23-31 in Figure S3: The 'old model' is replaced with 'initial version'; Line 29: The 'shallow green' & 'shallow blue' are replaced by 'shallow-green' & 'shallow-blue'; Line 30: 'The grey dotted vertical line in each plot represents zero on the y-axis.' is added.
- (2) Lines 35-42 in Figure S4: The '(orange dots)' & '(red dots)' are replaced with '(orange dots, ORC-D)' & '(red dots, ORC-AE)'; Line 37: The 'eddy covariance' is replaced by 'radiation'; Line 39: 'The black and blue solid lines are the simulated harvesting dates ( $T_{harvest}$ ) and the recorded tillage date ( $T_{tillage}$ ), respectively.' is deleted; Line 40-41: The 'shallow green area' & 'shallow blue area' are replaced by '(shallow-green area) ( $T_{yellowing, start}$ )' & '(shallow-blue area) ( $T_{res, end}$ )'; and 'harvesting date' is replaced with 'harvesting date ( $T_{harvest}$ )'.
- (3) Lines 45-52 in Figure S5: The 'old model' is replaced with 'initial version'; '[...]. The grey dotted vertical line in each plot represents zero on the y-axis.' is added.
- (4) Lines 61 in Figure S6: 'The grey dotted vertical line in each plot represents zero on the y-axis.' is added.
- (5) Lines 65-73 in Figure S7: 'Daily changes of total soil water content (SWC) (a) and SWC in different layers up to 2 m (b) in ORC-A' is replaced with '(a) Daily changes of total soil water content (SWC) in ORC-A'; Line 70-72: 'The grey dotted vertical line in each plot represents zero on the y-axis.' and '(b) Daily changes of SWC in different soil layers up to 2 m in ORC-AE compared to the initial version averaged over the harvested years across the twelve sites.' are added.
- (6) Lines 76-85 in Figure S8: The 'old model' is replaced with 'initial version'; the 'ORC-AE extends these modifications by adjusting surface albedo' is replaced with 'ORC-AE adjusts surface albedo'.
- (7) Lines 88-95 in Figure S9: 'The grey horizontal dotted line in each plot represents zero on the y-axis.' is added.
- (8) Lines 98-106 in Figure S10: The 'ORC-AE (green boxes) extends these modifications by adjusting  $\alpha_{surf}$ ' is replaced with 'ORC-AE adjusts  $\alpha_{surf}$ '; Line 101: The 'old model' is replaced with 'initial version'; Line 105: 'The grey horizontal dotted line in each plot represents zero on the y-axis.' is added.
- (9) Lines 109-117 in Figure S11: The 'ORC-AE (green boxes) extends these modifications by adjusting  $\alpha_{surf}$ ' is replaced with 'ORC-AE adjusts  $\alpha_{surf}$ '; Line 112: The 'old model' is replaced with 'initial version'; Line 116: 'The grey horizontal dotted line in each plot represents zero on the y-axis.' is added.
- (10) Lines 120-130 in Figure S12: 'The daily difference of latent and sensible heat flux (LE and H) between model simulations and observations across different soil water content intervals (SWC, %) during residue covering periods and the subsequent bare soil periods at 4 sites. ORC-AE (R-new and BS-new) adjusts surface albedo ( $\alpha_{surf}$ ), soil resistance ( $\beta_d$ ) and surface roughness ( $Z_0$ ) together in the ORCHIDEE-CROP model. ORC-D (R-old and BS-old) is the initial version. Observations were obtained from daily eddy-covariance measurements via the Integrated Carbon Observation System (ICOS) Data Portal. The periods of residue covering and bare soil were extracted by identifying site photos. 'R', 'BS' and 'OSS' in each plot

represent residues, bare soil and observation. '; Line 130: 'The mixing boxes for SWC intervals mean no data' is added.

Overall, the work is solid and novel. I recommend minor revision before acceptance.

**Reference**

Kozak, J. A., Ahuja, L. R., Green, T. R., and Ma, L.: Modelling crop canopy and residue rainfall interception effects on soil hydrological components for semi-arid agriculture, Hydrol. Process., 21, 229-241, https://doi.org/10.1002/hyp.6235, 2007.

Mitchell, J. P., Singh, P. N., Wallender, W. W., Munk, D. S., Wroble, J. F., Horwath, W. R., and Scow, K. M.: No-tillage and high-residue practices reduce soil water evaporation, California Agriculture, 66 (2), https://doi.org/10.3733/ca.v066n02p55, 2012.

Swella, G. B., Ward, P. R., Siddique, K. H. M., and Flower, K. C.: Combinations of tall standing and horizontal residue affect soil water dynamics in rainfed conservation agriculture systems, Soil and Tillage Research, 147, 30-38, <a href="https://doi.org/10.1016/j.still.2014.11.004">https://doi.org/10.1016/j.still.2014.11.004</a>, 2015.

Thapa, R., Tully, K. L., Cabrera, M., Dann, C., Schomberg, H. H., Timlin, D., Gaskin, J., Reberg-Horton, C., and Mirsky, S. B.: Cover crop residue moisture content controls diurnal variations in surface residue decomposition, Agricultural and Forest Meteorology, 308-309, 108537, <a href="https://doi.org/10.1016/j.agrformet.2021.108537">https://doi.org/10.1016/j.agrformet.2021.108537</a>, 2021.